# Conversion of invisible metal-organic frameworks to luminescent perovskite nanocrystals for confidential information encryption and decryption

Congyang Zhang[1], Bo Wang[1], Wanbin Li[2], Shouqiang Huang[1], Long Kong[1], Zhichun Li[1] & Liang Li[1]

Traditional smart fluorescent materials, which have been attracting increasing interest for security protection, are usually visible under either ambient or UV light, making them adverse to the potential application of confidential information protection. Herein, we report an approach to realize confidential information protection and storage based on the conversion of lead-based metal-organic frameworks (MOFs) to luminescent perovskite nanocrystals (NCs). Owing to the invisible and controlled printable characteristics of lead-based MOFs, confidential information can be recorded and encrypted by MOF patterns, which cannot be read through common decryption methods. Through our conversion strategy, highly luminescent perovskite NCs can be formed quickly and simply by using a halide salt trigger that reacts with the MOF, thus promoting effective information decryption. Finally, through polar solvents impregnation and halide salt conversion, the luminescence of the perovskite NCs can be quenched and recovered, leading to reversible on/off switching of the luminescence signal for multiple information encryption and decryption processes.

---

[1] School of Environmental Science and Engineering, Shanghai Jiao Tong University, 800 Dongchuan Road, 200240 Shanghai, People's Republic of China. [2] School of Environment, Guangzhou Key Laboratory of Environmental Exposure and Health, and Guangdong Key Laboratory of Environmental Pollution and Health, Jinan University, 601 West Huangpu Road, 510632 Guangzhou, People's Republic of China. Correspondence and requests for materials should be addressed to L.L. (email: liangli117@sjtu.edu.cn)

The development of stimuli-responsive fluorescent materials has attracted particular attention due to their potential security protection applications such as information storage, encryption, and anti-counterfeiting[1, 2]. With some external stimuli, the luminescent outputs of these materials can be tactfully changed, preventing the information or data from being stolen, mimicked, or forged. In the past decades, a range of smart luminescent materials including transition-metal complexes[3–5], organic dyes[6], inorganic semiconductor nanocrystals (NCs)[7–9], carbon dots[10–12], and lanthanide-doped upconverting nanoparticles[13–15] with changeable luminescent outputs have been explored. Through controlling their chemical constitutions or structures in the solid state, tunable luminescent signals can be achieved. These smart materials, however, suffer from many drawbacks such as insufficient luminescent performances, high material cost and/or tedious synthesis, and purification procedures. More importantly, due to their photoluminescent property, the data or informations recorded directly by these materials are usually visible under either ambient or UV light, which is adverse to their practical application for confidential information protection[16]. Therefore, it remains a great challenge to develop alternative cost-effective high-quality luminescent materials and systems with good confidential encryption abilities for high-level information storage and security protection applications.

Recently, metal halide perovskite $ABX_3$ (A = monovalent cations, B = divalent metal, and X = Cl, Br, I) materials have attracted great scientific interest in many research fields due to their excellent optoelectronic properties[17–19]. As fluorescent materials, low-cost lead halide perovskite nanomaterials exhibit bright photoluminescence (PL), relatively high PL quantum yields (PLQY), narrow emission spectra, and wide color gamut[20–24]. All the above advantages may make them excellent candidates for constructing smart luminescent systems[25]. However, direct use of perovskite nanomaterials as smart fluorescent systems is not a good choice because they are still visible under ambient or UV light, just like other fluorescent materials.

Herein, we propose an approach to realize confidential information protection and storage based on the conversion of lead-based metal-organic frameworks (MOFs) to luminescent perovskite NCs. As a relatively new class of porous hybrid materials, MOFs consist of inorganic metal centers and organic linkers by coordination bonds and possess a series of unique characteristics, including structure diversity, tunable pore sizes, tailorable functionality, and large surface area[26–29]. In our strategy, by employing MOFs as the lead source and the sacrificial porous template simultaneously, bright luminescent perovskite NCs in MOFs can be obtained via a direct conversion process simply triggered by a halide salt. Notably, the invisible and controlled

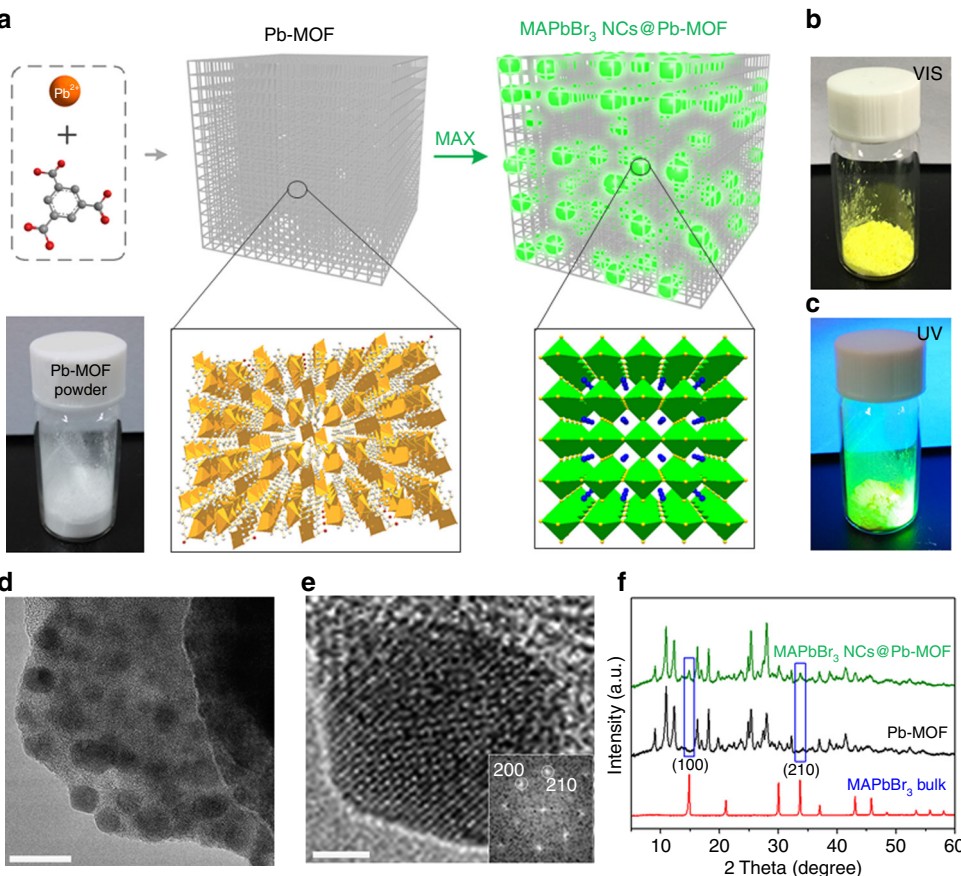

**Fig. 1** Conversion of a Pb-MOF to luminescent MAPbBr₃ NCs@Pb-MOF. **a** Schematic of the conversion process. MAX represents the halide salt (CH₃NH₃X, X = Cl, Br, or I). The green spheres in the matrix represent the MAPbBr₃ NCs. The two black boxes show 3D crystal structure of the Pb-MOF (left) and MAPbBr₃ (right). The Pb coordination polyhedra of the Pb-MOF (the Pb atom are coordinated by nine O atoms, in which two O atoms of one carboxylate coordinate to a Pb and also bridge two adjacent Pb atoms) and MAPbBr₃ are represented in orange and green, respectively. Other atom color scheme: C = gray, O = red, N = blue, Br = yellow. H-atoms have been omitted for clarity. **b, c** Optical images of MAPbBr₃ NCs@Pb-MOF powder under **b** ambient light and **c** 365 nm UV lamp; **d–f** Characterization of the MAPbBr₃ NCs@Pb-MOF: **d** TEM image, **e** HR-TEM image of one individual NC with the corresponding fast Fourier transformation image in the bottom right corner and **f** XRD patterns. Scale bar, 20 nm (**d**); 5 nm (**e**)

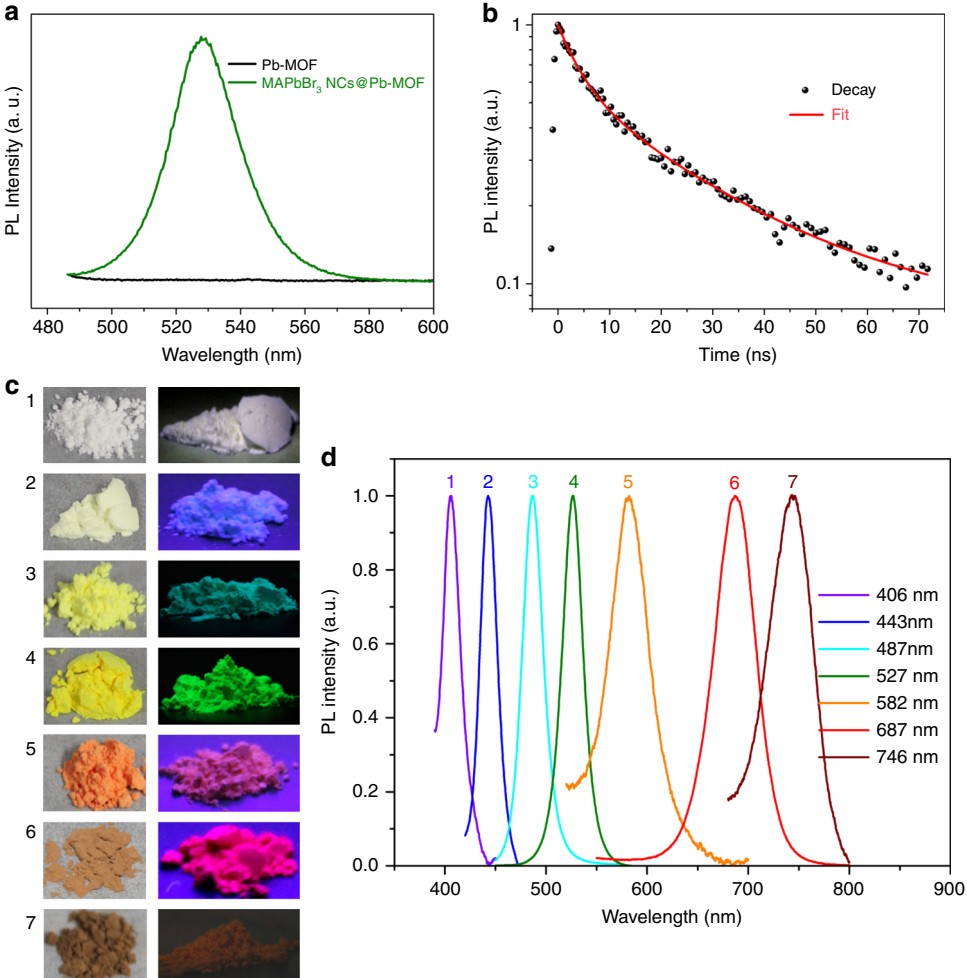

**Fig. 2** Optical properties of MAPbX₃ NCs@Pb-MOF. **a** Steady-state PL emission spectra of Pb-MOF (black line) and MAPbBr₃ NCs@Pb-MOF (green line). **b** Time-resolved PL decay curve of MAPbBr₃ NCs@Pb-MOF detected at 527 nm with excitation of 450 nm. **c** Optical images under ambient light and 365 nm UV lamp and **d** steady-state PL emission spectra of MAPbX₃ NCs@Pb-MOF. (1: MAPbCl₃, 2: MAPbCl₂Br, 3: MAPbClBr₂, 4: MAPbBr₃, 5: MAPbBr₂I, 6: MAPbBrI₂, 7: MAPbI₃)

printable characteristics of the lead-based MOFs allow us to easily record confidential information and protect them from general decryption methods (such as photochromic methods[30]). On the basis of the conversion strategy mentioned above, we successfully and easily realized the confidential information encryption and decryption process with various inkjet-printed patterns by using the invisible and stable Pb-MOF precursor as a security ink. In addition, due to the inherent ionic structure, the perovskite NCs in MOF matrix can be destroyed by polar solvents impregnation, thus quenching the luminescence of the perovskite NCs and realizing the recovery or even reversible on/off switching of the luminescence signal for multiple information encryption and decryption cycles.

## Results

**Conversion process**. Firstly, we demonstrated the feasibility of the conversion process and fabricated the $CH_3NH_3PbBr_3$ (MAPbBr₃) NCs from a lead-based MOF powder for a typical example. A lead-based MOF $(Pb_2(1,3,5-HBTC)_2(H_2O)_4$, Pb-MOF) was firstly synthesized according to a previous report (Fig. 1a)[31]. In this framework, $Pb^{2+}$ is coordinated by 1,3,5-HBTC²⁻ and two H₂O molecules with a square-pyramidal coordination geometry, resulting a 2D polymeric structure (Supplementary Fig. 1). Figure 1a schematically illustrates the

formation process of the MAPbBr₃ NCs in Pb-MOF matrix (named as MAPbBr₃ NCs@Pb-MOF). Different from the confined synthesis of pervoskite NCs in porous materials in recent reports[32–36], our strategy is based on a direct conversion process triggered by small amount of *n*-butanol solution containing $CH_3NH_3Br$ (MABr). To better control the conversion process, we used hexane to disperse the MOF powder. The emission color of the suspension changed quickly from non-fluorescence to blue green then to yellow green (Supplementary Fig. 2), indicating an obvious quantum-confinement phenomenon due to the growth of the perovskite NCs. After several minutes, the bright powder (Fig. 1b) was collected by filtration, rinsed, and dried. As shown in Fig. 1c, the resulting MAPbBr₃ NCs@Pb-MOF powder shows brilliant green emissions under a UV lamp (365 nm).

**Morphology and structure characterization**. Figure 1d, e and Supplementary Fig. 3 show the transmission electron microscopy (TEM) and scanning electron microscopy (SEM) images of the MAPbBr₃ NCs@Pb-MOF sample, respectively. As shown in TEM image (Fig. 1d), the synthesized MAPbBr₃ NCs are well distributed in Pb-MOF with a diameter of about 10–20 nm. From the high-resolution transmission electron microscopy (HR-TEM) and the fast Fourier transformation (FFT) images (Fig. 1e), the interplanar distances of 2.98 and 2.62 Å, corresponding to the

(200) and (210) crystal faces of the cubic MAPbBr$_3$ crystal, respectively, can be easily confirmed[21, 37]. The SEM images (Supplementary Fig. 3) show that the surface of Pb-MOF crystals become rough after the conversion process. In addition, crystal structure was characterized by X-ray diffraction (XRD) and shown in Fig. 1f. The XRD pattern of Pb-MOF is similar to the literature[31]. After conversion, the framework structure of Pb-MOF is well retained. Apparently, we can preliminary confirm the existence of MAPbBr$_3$ NCs in Pb-MOF from the two new peaks at 14.9° and 33.7° correspond to (100) and (210) planes of the cubic MAPbBr$_3$ (space group: Pm3m No. 211). To further demonstrate the conversion process, on one hand, the Pb-MOF and MAPbBr$_3$ NCs@Pb-MOF were analyzed by X-ray photo-electron spectroscopy (XPS). As shown in the full-range XPS spectra (Supplementary Fig. 4), the new signals of Br, N species appear distinctly. For more detail analysis, compared to the Pb-MOF, the Pb 4$f$ peaks shift to lower binding energies (BE) (from 139.0 to 138.7 eV) and slightly broaden for the MAPbBr$_3$ NCs@Pb-MOF sample, suggesting the change of coordination chemistry of Pb atoms (the emerging Pb–Br bond from MAPbBr$_3$ NCs). On the other hand, a long time (48 h) and high reaction concentration of MABr (20 times than used for luminescent MAPbBr$_3$ NCs@Pb-MOF) conversion process was conducted to consume almost all of the Pb from MOFs, which confirmed by the XRD characterization. As shown in Supplementary Fig. 5, the diffraction signals of Pb-MOF framework have markedly reduced, whereas the sharp peaks of MAPbBr$_3$ are shown predominately, which indicates that the Pb elements for MAPbBr$_3$ NCs are indeed from Pb-MOF rather than the residual Pb$^{2+}$ in MOF matrix. In general, all of the above results have robustly demonstrated the successful conversion process of Pb-MOF to MAPbBr$_3$ NCs. Furthermore, the percentage of the perovskite NCs in MOF matrix was estimated by XRF (only ~3%, Supplementary Table 1) and XPS analysis (~19%, Supplementary Table 2). Notably, the difference between these results (3% vs. 19%) can be ascribed to the different detecting depths of the two analysis methods. Compared with XRF, the detecting depth of XPS is only several nanometers[38, 39], which may suggest that perovskite NCs are mainly located in the outer part of MOF particle. Also by employing the MOFs as self-templates, a series of nanomaterials (such as metal sulfide: CuS[40], ZnS[41], and metal oxides: ZnO[42] et al.) have been synthesized in previous reports. Accordingly, the formation mechanism of the MAPbX$_3$ NCs in Pb-MOFs in our work could be attributed to the synergetic roles of the dissolution rate of Pb$^{2+}$ from the MOF, the diffusion rate of MA$^+$ and X$^-$ toward the Pb-MOF and the formation rate of MAPbX$_3$ NCs, in which the formation rate is larger than the dissolution and diffusion rates[41].

**Optical characterization**. Figure 2a shows the PL spectra of as-synthesized MAPbBr$_3$ NCs@Pb-MOF and Pb-MOF powder. Obviously, Pb-MOF does not show any florescence signal in the visible range. But the MAPbBr$_3$ NCs@Pb-MOF exhibits a green emission peak at 527 nm with narrow 25 nm full-width-at-half-maximum (FWHM). The relative sharp emission highlights the outstanding superiority of the luminescent perovskite NCs over the traditional smart fluorescence materials. The excitation-emission matrix (EEM) spectrum of the MAPbBr$_3$ NCs@Pb-MOF powder, shown in Supplementary Fig. 6, reveals that the PL emission is not excitation wavelength dependent. The UV–vis absorption spectra (Supplementary Fig. 7) show that MAPbBr$_3$ NCs@Pb-MOF exhibits a broad absorption at 350–550 nm. In contrast, the Pb-MOF also does not show any absorption signal in the visible range, thus further indicating the "invisible" characteristic of the Pb-MOF. "Invisible" is in quotation marks

because the Pb-MOF powder here, showing white color (Fig. 1b), is still visible under ambient light due to the existing scattering phenomenon. We should note that the PL peak or excitonic absorption peak wavelength of our bright powder sample is larger than the reported MAPbBr$_3$ quantum dots (QDs)[21, 35], which can be attributed to the large crystal size compared to the small excitonic Bohr radius for MAPbBr$_3$ (~1.4–2 nm)[43]. We hold that it is the final rinsing or drying step rather than conversion process that leading to the relative large crystal size of our perovskite sample because the quantum-confinement phenomenon can be obviously observed during the perovskite NCs' growth process (Supplementary Figs. 2 and 8). In addition, the absolute PLQY of the as-synthesis MAPbBr$_3$ NCs@Pb-MOF powder is 39.6% determined by a fluorescence spectrometer with an integrated sphere executed at a wavelength of 390 nm. Compared to the luminescent perovskite NCs synthesized by conventional solution-processable strategies initially outlined by Pérez-Prieto et al.[20], the relatively lower PLQYs of the MAPbBr$_3$ NCs@Pb-MOF can be attributed to the absence of any surface shelling and insufficient ligand passivation. In spite of this, it is comparable to or even brighter than these reported MAPbBr$_3$ NCs confined synthesized in porous matrix[32–36], and sufficient for information identification applications. The time-resolved PL spectrum is shown in Fig. 2b. The PL decay can be described by biexponential fitting, giving a short-lived PL lifetime ($\tau_1$) of 4.4 ns with a percentage of 44.3% and long-lived PL lifetime ($\tau_2$) of 26.2 ns with a percentage of 55.7%. Similar to the previous report[35], the shorter lifetime could be the result of dominant surface trapping of the MAPbBr$_3$ NCs, suggesting that the non-radiative recombination pathway has non-negligible contribution in our MAPbBr$_3$ NCs@Pb-MOF sample, which is consistent with the above-mentioned result of relative low PLQY. Moreover, we have demonstrated that the conversion process also can be applied to the CsPbX$_3$ NCs. As a representative, the CsPbBr$_3$ NCs has been fabricated via a similar conversion process from the Pb-MOF. The optical properties and photograph of CsPbBr$_3$ NCs@Pb-MOF are shown in Supplementary Fig. 9, which indicates the versatility of our conversion approach. In addition, by adjusting their halide composition (X = Cl, Br, and I), the emission color of as-synthesized MAPbX$_3$ NCs@Pb-MOF can be tuned over the entire visible spectral region. Figure 2c and d show the optical images (under ambient and UV lamp) and the PL emission spectra of a series of MAPbX$_3$ NCs@Pb-MOF samples with varied halide compositions, which has been easily tuned from deep blue to near infrared with relative narrow emissions (FWHM = 19–55 nm). From the XRD characterization (Supplementary Fig. 10), the peaks of the (100) and (210) reflection gradually shift toward higher angles with smaller halide ions (Br, Cl) due to the reduced lattice parameters, which confirm the cubic perovskite phase for all MAPbX$_3$ NCs samples.

**Confidential information encryption and decryption application**. Benefiting from the above "invisible" advantage of the Pb-MOF and the robust conversion strategy of luminescent perovskite NCs, our perovskite NCs-MOFs platform may has great potential to realize the confidential information encryption and decryption process. Moreover, inspired by previous reports, the precise control of positioning and patterning of MOFs with high-resolution can be easily realized[44–46], which offers another advantage of our platform for large area and high-density printable applications[47, 48]. Among numerous patterning technologies, inkjet printing is particularly attractive because of the mask-free, high-spatial resolution, and continuous operation advantages[49–51]. In this manuscript, an invisible and stable precursor solution of Pb-MOF has been used as the security ink directly to

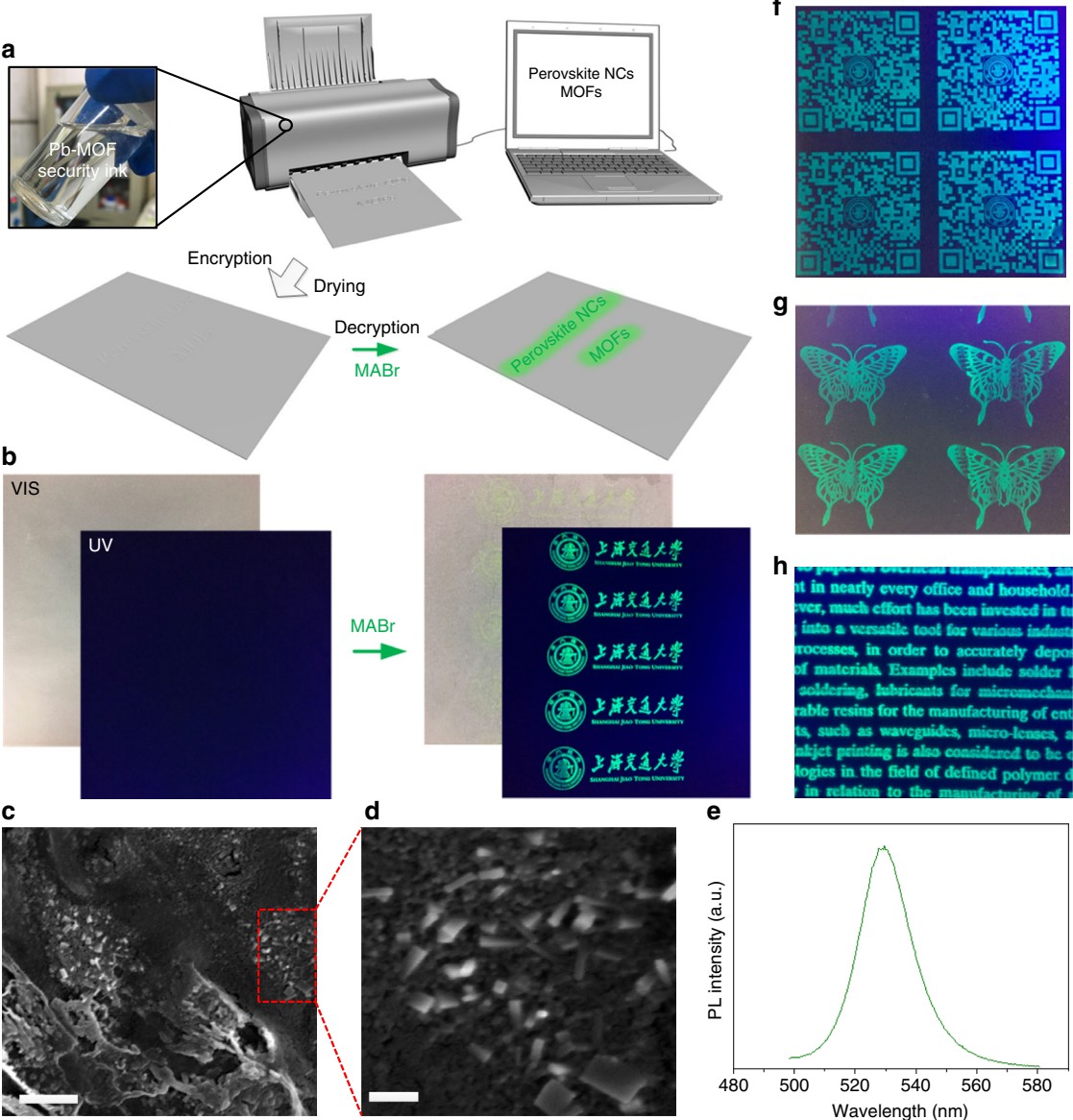

**Fig. 3** Luminescent perovskite NC patterns from a Pb-MOF via inkjet printing. **a** Schematic illustrations of the patterning, information encryption, and decryption process of the perovskite NCs-MOF platform. **b** Digital images of the printed logo of Shanghai Jiao Tong University on a commercial parchment paper before and after MABr loading under ambient light and a 365 nm UV lamp. **c, d** SEM images of the parchment substrates with Pb-MOF pattern. Scale bar, 1 μm (**c**); 200 nm (**d**). **e** PL spectrum of the MAPbBr$_3$ NCs@Pb-MOF pattern on parchment substrate. **f–h** Printed complicated patterns: QR code, butterfly, and characters, respectively

print various patterns by an inkjet printer. It is worth mentioning that the viscosity and surface tension of the ink have important role for the inkjet-printing process. Therefore, inspired by Zhuang's report[52], a combinational solvent system containing dimethylsulfoxide (DMSO), ethanol, and ethylene glycol (EG) has been employed to prepare the Pb-MOF precursor solution. Figure 3a illustrates the patterning, information encryption, and decryption process of our perovskite NCs-MOF platform. Through the inkjet-printing process, MOF precursor can be easily deposited onto desired positions using a nozzle. After the solvent evaporated by the drying step, small MOF crystals would nucleate and grow in specific areas. Notably, the printed Pb-MOF nanoscale crystals and the "invisible" characteristic of the Pb-MOF jointly promote the absolutely and really invisible characteristic of Pb-MOF patterns because of the reduced or even negligible scattering[53–55], which is significant and necessary for confidential information encryption. Then the information

decryption process can be conducted by conversion reaction of Pb-MOF crystal on substrate via loading *n*-butanol solution containing MABr by a sprayer. About several minutes later, the solvent has been evaporated and the bright green emission pattern appeared clearly under UV light excitation.

Figure 3b illustrates the printed logo of Shanghai Jiao Tong University on a commercial parchment paper. Obviously, the printed Pb-MOF pattern is indeed invisible absolutely. The XRD (Supplementary Fig. 11) shows the crystalline characteristics of the above-mentioned Pb-MOF with a partial preferential orientation mainly in the (001), (100), (101) directions. The SEM images of the printed Pb-MOF pattern, shown in Fig. 3c and d, suggests that the Pb-MOF pattern is composed of many nanoscale crystals (about 200–400 nm). For the security protection applications, the stability of MOF seems to be a critical factor. As shown in Supplementary Fig. 12a, the thermogravimetry analysis (TGA) shows that the as-synthesized Pb-MOF is

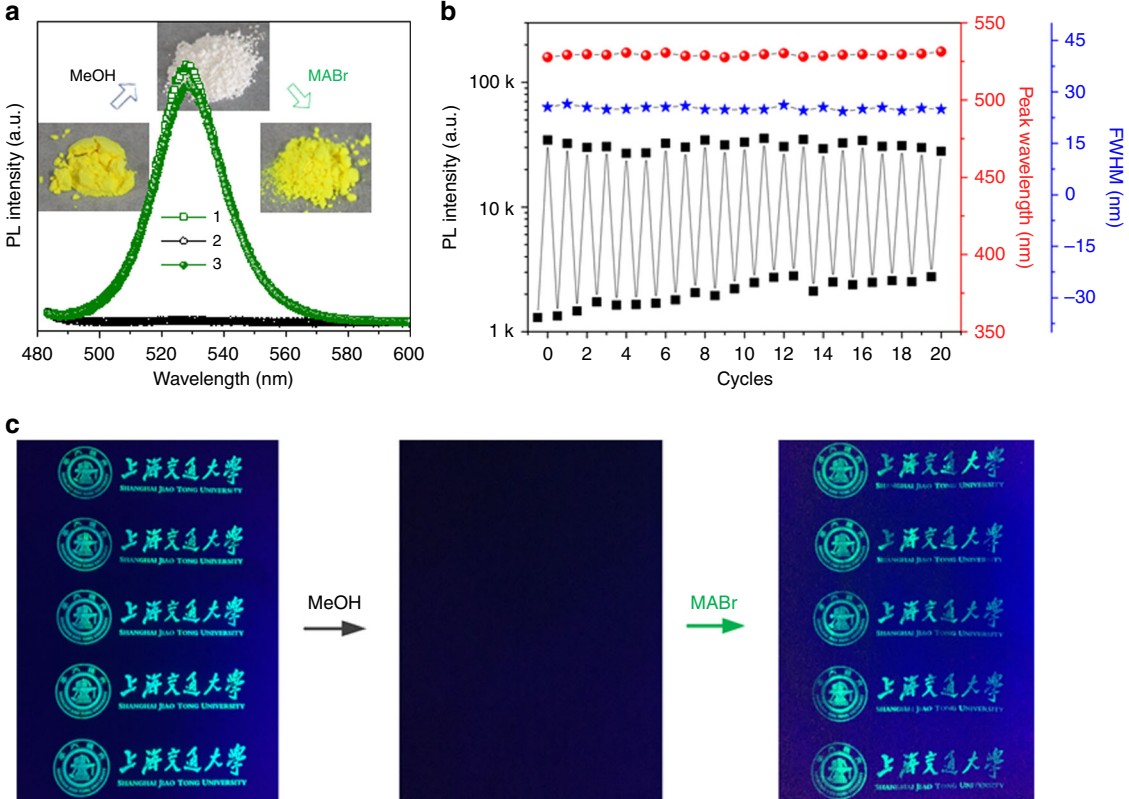

**Fig. 4** The reversible on/off switching property of perovskite NCs-MOF platform. **a** Sequential optical images and PL emission spectra of MAPbBr$_3$ NCs@Pb-MOF after one cycles of impregnation-recovery process. 1, 2, and 3 represent the original, impregnated, and recovered powder sample of MAPbBr$_3$ NCs@Pb-MOF, respectively. **b** PL intensity, peak wavelength, and FWHM of MAPbBr$_3$ NCs@Pb-MOF in the impregnation-recovery cycles as a function of cycle number. **c** Reversible fluorescence switching of the MAPbBr$_3$ NCs@Pb-MOF pattern in one encryption–decryption cycle (methanol impregnation for encryption and MABr spraying for decryption)

stable to 400 °C indicating that it has a good thermal stability, which is consistent with previous report[31]. On the other hand, from the XRD characterization (Supplementary Fig. 12b, c), it is obviously that both the Pb-MOF powder and the printed Pb-MOF pattern can remain original crystal structure after storage of several months, thus suggesting the excellent storage stability. After loading of MABr, the logo comes out very pale yellow green color under ambient light and bright green color under UV light illumination. To be sure, benefiting from the excellent fluorescent properties of the perovskite NCs (high PLQY, sharp emission), the almost colorless and invisible MAPbBr$_3$ NCs@Pb-MOF pattern under ambient light also could be obtained by using MOF precursor with extremely low concentration without affecting the decryption process. The PL spectrum (Fig. 3e) reveals that the MAPbBr$_3$ NCs@Pb-MOF pattern on parchment paper has a narrow emission peak at 529 nm, similar to the above bright powder sample. From the XRD characterization of the printed MAPbBr$_3$ NCs@Pb-MOF pattern in Supplementary Fig. 11, three typical diffraction peaks of cubic MAPbBr$_3$ appear obviously. Moreover, as shown in Fig. 3f–h, various complicated patterns (including QR code, butterfly, and characters) have also been printed with good resolution. To demonstrate the necessity and the role of the MOF structure in the security protection application of our platform, the corresponding Pb$^{2+}$ ink (without H$_3$BTC linker) was prepared and used for information encryption and decryption process. As shown in Supplementary Fig. 13, it is obvious that the printed pattern using Pb$^{2+}$ ink cannot maintain the information encryption and decryption capability on substrates. The mechanical properties of the Pb-MOF and MAPbBr$_3$ NCs@Pb-MOF pattern were also qualitatively assessed

through a typical tape peel test[56, 57]. As shown in Supplementary Fig. 14, after tape adhesion and peeling, the Pb-MOF and MAPbBr$_3$ NCs@Pb-MOF pattern both can kept their high quality, which indicates that printed Pb-MOF and MAPbBr$_3$ NCs@Pb-MOF materials have excellent mechanical stability for security protection applications. Importantly, we also find that the obtained MAPbBr$_3$ NCs@Pb-MOF pattern on parchment paper also exhibits good stability stored in air. As shown in Supplementary Fig. 15, the MAPbBr$_3$ NCs@Pb-MOF pattern exposed in air after 3 months is invisible under ambient light but still exhibits green emission under the UV illumination. This phenomenon can be ascribed to the good protection effect of the textured substrate and the MOF matrix[36].

**Universality of perovskite NCs-MOF platform.** Apart from parchment paper, the patterns also can be printed on transparent PET foils as well (Supplementary Fig. 16), enabling our perovskite NCs-MOF system's great potential for promising applications on multi-integrated light sources or other optoelectronic devices[58, 59]. In addition, the unique color tunable property of luminescent perovskite materials allow us to change the emission color of our patterns via replacing MABr with other halide salt. Supplementary Figure 17 illustrates the deep red colored patterns using an iodine-containing chromogenic reagent. Because of the relatively lower PLQY of the iodine-containing perovskite NCs, the deep red colored pattern is also a little bit dull. To further present the versatility of this strategy, as shown in Supplementary Fig. 18, we printed the letters of SJTU (the acronym of Shanghai Jiao Tong University) on parchment paper via contact-printing technique[60]. For more practical application, our

perovskite NCs-MOF platform also shows good performance for anti-counterfeiting application on banknotes (Supplementary Fig. 19).

**Reversible on/off switching of luminescence signals.** Owing to the inherent ionic structure, the pervoskite framework is vulnerable and can be destroyed easily by many harsh conditions[61–65]. Inspired by this unique property, herein, we find that the MAPbBr$_3$ NCs in Pb-MOF can be destroyed by polar solvents impregnation (e.g., methanol), enabling the quenching of the luminescence of perovskite NCs and may realizing the reversible on/off switch of the luminescence signal for multiple information encryption and decryption processes. Figure 4a displays the photographs and the PL emission spectra of the MAPbBr$_3$ NCs@Pb-MOF powder samples within one cycle of impregnation-recovery process. After methanol impregnation and reaction with MABr again, the color of the as-synthesized powder changes to white and backs to yellow green. The as-synthesized MAPbBr$_3$ NCs@Pb-MOF powder exhibits a strong fluorescence peak centered at 528 nm (curve 1). After impregnation and rinsing with methanol, the fluorescence of the sample is markedly quenched: the PL intensity decreases to only 0.4% (curve 2). Further, when the discolored powder reacted with MABr again, the fluorescence recovered and the PL intensity reaches 93.8% of the original value (curve 3). From the XRD and Fourier transform infrared (FTIR) data (Supplementary Fig. 20a, b) of the MAPbBr$_3$ NCs@Pb-MOF powder sample after methanol impregnation, the distinct diffraction peak at 14.9° and the weak absorption peak at 977 cm$^{-1}$[66] both disappear quickly. Furthermore, the XPS characterization of the MAPbBr$_3$ NCs@Pb-MOF powder, shown in Supplementary Fig. 20c and d, clearly reveals that after degradation, the signal of Br significantly reduced, whereas the N species almost disappeared. All of these data suggest that the luminescent MAPbBr$_3$ phase can been degraded by the methanol treatment. Considering the small amount of the perovskite NCs and the complex hybrid system, it is difficult to determine the accurate degradation products from these results. To further figure out the degradation pathway, the methanol impregnation experiment of MAPbBr$_3$ bulk sample has been directly conducted to avoid the influence of the MOF matrix (the Pb-MOF does not change or degrade in methanol). The results (including XRD and FTIR), shown in Supplementary Fig. 21, demonstrate that after methanol impregnation, the MAPbBr$_3$ NCs may mainly decompose into PbBr$_2$ and other C, N-containing organic constituents, which can be easily washed away because of their good solubility (in methanol) and volatility. However, based on the small amount of MAPbBr$_3$ NCs in Pb-MOF, the Pb element from the destroyed MAPbBr$_3$ NCs may have a relatively small role for the next conversion process. In this regard, the Pb-MOF actually acts as a huge reservoir of metal source for the repeated formation of luminescent perovskite NCs, which allows the reversible on/off switching with high-quality fluorescent property of our platform. To further examine the reversible property, 20 cycles were conducted. As shown in Fig. 4b, negligible decrease in PL intensity is observed after 20 consecutive switching cycles. In addition, the peak wavelength and FWHM of the PL spectra almost remain the same. Similar to the powder, the luminescence of MAPbBr$_3$ NCs@Pb-MOF pattern can also be quickly quenched by the methanol impregnation, and recovered again by MABr loading with high quality (Fig. 4c). Moreover, even after 10 consecutive switching cycles, the quality of the information encryption and decryption process still remains unaffected (Supplementary Fig. 22), which indicates that our platform can be applied for multiple information encryption and decryption processes.

## Discussion

In summary, we have developed an approach to realize confidential information protection based on the conversion of invisible Pb-based MOFs to luminescent perovskite NCs. Through simply reacting with CH$_3$NH$_3$X (MAX, X = Cl, Br, or I) salt, the MAPbX$_3$ NCs with bright luminescence can be rapidly obtained in a Pb-MOF matrix. Owing to the invisible and controlled printable characteristic of the Pb-MOF and the conversion process of luminescent perovskite NCs, we have demonstrated that our platform can act as a smart luminescent system towards confidential information encryption and decryption with various high-resolution patterns by inkjet-printing technique, which protect the recorded confidential information from general decryption methods. In addition, the inherent ionic structure of the perovskite materials allows us to quench the luminescence of the as-synthesized perovskite NCs easily by polar solvent (methanol) impregnation and realize the reversible on/off switching of the luminescence signal for multiple information encryption and decryption processes. It should be mentioned that the toxicity of Pb in lead halide perovskites[67–69] is a considerable concern for the future applications of our platform. However, it would not be a fatal problem according to the recent fascinating research progresses in the synthetic chemistry of lead-free perovskite materials. Jellicoe et al.[70] and Wang et al.[71] have both reported successful replacement of lead with non-toxic tin by synthesizing CsSnX$_3$ and Cs$_2$SnI$_6$ NCs, respectively. In addition, Leng et al.[72] prepared a novel MA$_3$Bi$_2$Br$_9$ QDs with high PLQY, suggesting that bismuth is another promising choice for toxic metal-free perovskite materials. We believe that our strategy will open up a potential avenue for luminescent perovskite materials in security-protecting applications.

## Methods

**Materials.** Lead nitrate (Pb(NO$_3$)$_2$, 99%), trimesic acid (1,3,5-H$_3$BTC, 98%), methylamine (CH$_3$NH$_2$, 30–33 wt.% in methanol solution), hydrobromic acid (HBr, 48% in water), hydroiodic acid (HI, 57% in water), cesium bromide (CsBr, 99.9%), cesium iodide (CsI, 99.9%), dimethylsulfoxide (DMSO, >99%) were acquired from Aladdin. Butanol (≥99.5%), methanol (≥99.5%), n-hexane (≥97%), toluene (≥99.5%), ethanol (≥99.7%), ethylene glycol (EG, ≥99%), acetic acid (HCl, ≥99.5%) were purchased from Sinopharm Chemical Reagent Co. Ltd, China.

**Synthesis of Pb-MOF.** The Pb-MOF ([Pb$_2$(1,3,5-H$_3$BTC)$_2$(H$_2$O)$_4$]·H$_2$O) was synthesized following the previous procedure reported by Sadeghzadeh with some modification[31]. Briefly, 90 mL of a 0.01 M solution of 1,3,5-H$_3$BTC in water was positioned in a high-density ultrasonic cleaner and 10 mL of a 0.09 M Pb(NO$_3$)$_2$ in water was added dropwise to the solution at room temperature. After addition, the solution remained in the bath for 30 min. The obtained precipitates were filtered, washed with ethanol, and then dried.

**Synthesis of MAPbX$_3$ NCs@Pb-MOF and CsPbX$_3$ NCs@Pb-MOF powder.** The MAX (X = I, Br, Cl) were synthesized according to our previous reports[65]. Briefly, CH$_3$NH$_3$X (X = Cl, Br, or I) was synthesized by the reaction of methylamine and HBr with the molar ratio of 1: 1. After reaction for 2 h, the precipitate was obtained by rotary evaporation, followed by washing with diethyl ether, and then dried under vacuum. For the synthesis of MAPbX$_3$ NCs@Pb-MOF, the Pb-MOF (200 mg) was dispersed in hexane (10 mL) with stirring at room temperature. For the typical synthesis of MAPbBr$_3$ NCs@Pb-MOF, the Pb-MOF (200 mg) was dispersed in hexane (10 mL) with stirring at room temperature. Then 500 μL MABr/n-butanol solution (10 mg/mL) was added to the Pb-MOF/hexane suspension. After 30 s, the colored powder was collected by filtration and rinsed with n-butanol then dried at 80 °C for 5 min before further characterization. MAPbBr$_x$Cl$_{3-x}$ NCs@Pb-MOF and MAPbBr$_x$I$_{3-x}$ NCs NCs@Pb-MOF were fabricated via the similar strategy with the variation of the MAX/n-butanol solution constitution. The synthesis of CsPbBr$_3$ NCs@Pb-MOF was similar to the MAPbBr$_3$ NCs@Pb-MOF. The difference was that the MABr/n-butanol solution (10 mg/mL) and hexane were replaced with CsBr/methanol solution (20 mg/mL) and toluene, respectively. For the quenching of MAPbBr$_3$ NCs@Pb-MOF, the MAPbBr$_3$ NCs@Pb-MOF powder was dispersed in methanol with stirring at 25 °C. After 10 min, the powder was collected by centrifugation and then dried at 80 °C.

**Characterization.** The powder XRD measurements were measured on a Bruker D8 Advance X-ray Diffractometer at 40 kV and 40 mA using Cu Kα radiation (λ =

1.5406 Å). The HR-TEM and scanning transmission electron microscope high-angle annular dark-field (STEM HAADF) images were obtained from the FEI Talos F200X TEM instruments operated at an accelerating voltage of 200 kV. The SEM images were obtained from the JEOL JSM-7800F field emission scanning electron microscope (FESEM). The chemical compositions were determined by the XPS (Kratos Axis Ultra$^{DLD}$, all the binding energies were calibrated with the C 1$s$ peak at 284.8 eV) and X-Ray Fluorescence measurement (XRF-1800 spectrometer equipped with a Rh anode X-ray tube). FTIR spectra were measured by using a Nicolet 6700 spectrometer (USA). TGA was conducted on a Mettler Toledo analyzer from 30 to 700 °C at a heating rate of 10 °C/min with an Ar flow rate of 50 mL/min. PL spectra were taken using a F-380 fluorescence spectrometer (Tianjin Gangdong Sci. & Tech. Development Co., Ltd., China). The total luminescence spectra were characterized in the form of EEM (F-7000, Hitachi) with the scanning emission spectra varied from 450 to 600 nm by increasing the excitation wavelength from 300 to 550 nm at 5 nm increments. The PL decay curves were recorded on a PTI QM/TM/IM fluorescence spectrofluorometer. The absolute PLQY were detected using a fluorescence spectrometer with an integrated sphere executed at the 390 nm LED light source.

**Security protection application**. The Pb-MOF ink solution was prepared by dissolving 2.15 g Pb(NO$_3$)$_2$ and 0.58 g 1,3,5-H$_3$BTC in 5 mL DMSO, then 11.25 mL ethanol (EtOH) and 7.5 mL ethylene glycol (EG) were added under vigorous stirring. Finally, the mixture was filtered through a 0.22 μm syringe filter. The security printing tests were performed on a modified HP inkjet printer (HP Desk Jet 2132) with customized ink cartridges. The customized cartridge was washed with H$_2$O and EtOH and dried in air. About 5 mL of Pb-MOF ink solution was loaded into the empty cartridge and then loaded the cartridge into the printer. After printing, the substrate was dried in ambient, and then immersed into methanol solution for 30 min for a solvent development process to remote the residual solvents and dried again. For the decryption process, the printed substrate with Pb-MOF patterns was loaded with MABr/$n$-butanol (10 mg/mL) solution by a sprayer. After dried in ambient for several minutes, the shallow yellow green pattern can be observed by the naked eye. Under UV light illumination, the patterns showed bright green emission clearly. For the further encryption process, the substrate with patterns was immersed in methanol solution for about 10 min, the green emission was gradually quenched and finally disappeared absolutely.

**Data availability**. All relevant data supporting the findings of this study are available from the corresponding authors on request.

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

## Acknowledgements

This work is supported by the National Natural Science Foundation of China (21271179, 21607101) and the Program for New Century Excellent Talents (NCET-13-0364).

## Author contributions

L.L. and C.Z. conceived the original research idea. Experiments, including Pb-MOF fabrication, $MAPbBr_3$ NCs@Pb-MOF synthesis, and inkjet-printing test were performed by C.Z., B.W., and W. L. C.Z., S.H., L.K., and Z.L. carried out characterizations and analyses including transmission electron microscope, scanning electron microscopy, powder diffraction, X-ray photoelectron spectroscopy, UV–vis absorption, PL, PL life-time, and PLQY. The manuscript was co-written by L.L. and C.Z. All authors contribute to the discussion and revising of this manuscript.
