## [Peer Review File · Nature Communications]

Reviewers' comments:

Reviewer #1 (Remarks to the Author):

The authors used a simple method to realize confidential information protection based on the reaction of MAX and Pb based MOF. They demonstrated that the confidential information can be recorded and encrypted by invisible MOFs patterns. Perovskite NCs will promote information decryption by bright luminescent after reacting with MOFs. Also, due to the inherent ionic structure of perovskite NCs, they can realize reversible on/off switch of luminescence for multiple information encryption and decryption processes. I think it is an important paper for security protecting applications. However, before it may be accepted for publication, several issues should be addressed.

1. The authors claim that the difference of amount of the perovskite NCs in MOF matrix estimated by EDX analysis of TEM (1%) and XPS data (19%) is due to the different detecting depths. However, the amount from the two results are quite different. Could the authors give some statistic data? Since MABr can only react with a portion of MOFs, they should carefully consider the uniformity of the samples.

2. In Figure 2c,d, the authors show MAPbX₃NCs@Pb-MOF can be tuned over the entire visible spectral region by adjusting their halide composition. What is the exact halide composition for each sample?

3. In the multiple information encryption and decryption processes, the authors found that the MAPbBr₃ NCs@Pb-MOF powder encounter obvious degradation. They guess the MAPbBr₃ NCs@Pb-MOF powder may degrade to some new compounds like MA₄PbBr₆·MeOH, MA_xPbBr_{2+x}, PbBr₂, Pb(OH)₂ or PbCO₃. I hope the authors give sufficient evidence for this assumption, such as Raman or FTIR, etc.

Reviewer #2 (Remarks to the Author):

The manuscript titled "Conversion of Invisible Metal-Organic Frameworks to Luminescent Perovskite Nanocrystals for Confidential Information Encryption and Decryption", by Liang Li and co-work describe a novel approach to realize confidential information protection and storage based on the conversion of Pb based metal-organic frameworks (MOFs) to luminescent perovskite nanocrystals. This approach provide high capability of encryption to transmit confidential information and very simple procedure for decryption of this. Before being published, some adjustments and experiments need to be performed, as follows.

- i) The authors need to provide informations of mechanical properties, through adhesion and adhesion experiments, of the printed and MAPbX₃ NCs@Pb-MOF materials printed;
- ii) If the Pb salt were printed in place of ink MOF, maintaining the concentrations of Pb and solvent, the incryption capability of would be maintained? The experiment to expose the Pb-MOF to MA-Br indicate only that the Pb ions of the MOF react as MA-Br, But does not eliminate the other hypothesis. The authors need to provide these information;
- iii) The degradation of MOF seems to be a limiting factor for the application of this approach to explore the anti-counterfeiting applications. The limitations must be related in the manuscript;
- iv) Provide adequate section title;
- v) Improve the legends of Figures 2 and 4, in order to facilitate the correlation with the materials;
- vi) Improve discussion of the Figure 2;
- vii) The sections name need be reviewed, specifically the introduction.

Therefore, I recommend accept with major corsctions.

Reviewer #3 (Remarks to the Author):

In this paper, the authors developed a new strategy to realize confidential information protection and storage based on the conversion of Pb-MOF to luminescent perovskite NCs. By employing MOF as the Pb source and the sacrificial porous template simultaneously, bright luminescent perovskite NCs in MOF could be obtained via a directly conversion process. Due to the simple conversion method and the invisible character of the Pb-MOF, the author have successfully demonstrated that the potential application of the confidential information encryption and decryption. The idea and the experimental results are interesting and impressive. Based on the novelty and quality of this manuscript, I think that this manuscript can be published in Nature Communications after the following revisions:

(1)The authors have emphasized luminescent perovskite NCs obtained from the conversion of the invisible Pb-MOF as an important potential application for confidential information encryption and decryption. Even though the stability of the perovskite NCs has been improved by the protection effect of the textured substrate and the MOF matrix, it seem still incompatible with the practical application in view of their toxic nature of the Pb²⁺ ion in the perovskite NCs. The authors are suggested to address this concern with more discussion or perspectives to tackle this technical challenge.

(2)In the section of "Optical characterization of MAPbX₃ NCs@Pb-MOF powder", the author considered that the Pb-MOF powder, showing white color (Figure 1b), is still visible under ambient light due to the existing scattering phenomenon. Then, in the section of "Confidential information encryption and decryption application", the authors claimed that the printed Pb-MOF pattern is indeed invisible absolutely because of the reduced or even negligible scattering phenomenon. This point is not enough clear, because the substrates used for the printing process seem also white color, it means you are printing white MOFs on a white substrate, which may confuse the observation, The author should give some references to support the standpoints of the existing scattering phenomenon of MOF powder and the reduced or even negligible scattering phenomenon through reduce the crystal size of MOF.

(3)More details should be added in certain Figures' caption, for example, what are the emission and excitation wavelengths of the decay spectrum in the time-resolved decay curve measurement in Fig. 2b; what the 7 samples in Fig. 2c and 2d are; how to treat the printed substrate by MeOH and MABr in Fig.4c.

(4)Abbreviations should be clarified when they are first used, for example, MABr is a critical reagent in this study, but I did not get what it is.

(5)The emission spectra of MAPbBr₃NCs@Pb-MOF under different excitation wavelengths should be measured.

(6)High resolution TEM and FFT images of the "small MAPbBr₃ QDs" should be applied to prove the same crystal structure with MAPbBr₃NCs in Pb-MOF.

(7)As to the encryption and decryption processes, the authors stated that "for the further encryption process, the substrate with patterns was immersed in methanol solution for about 10 min, the green emission was gradually quenched and finally disappeared absolutely", I think that the printed materials on the substrate should gradually dissolve in methanol, because the printed material is also dissolved in alcohols solution (i.e. ethanol and ethylene glycol).

(8)There are some sloppy statement and spelling mistakes in this manuscript. I listed out some of

them below.

- a. In the line of 77 ~ 78, "the emission color ... from white to blue green then to yellow green" should change into "the emission color ... from non-fluorescence to blue green then to yellow green" because the white color of MOF powder is not the emission color.
- b. In the line of 17, "quickly simply" should be "quickly and simply".
- c. In the line of 54, "could" should be "can".
- d. In the line of 86, "N=blue, Br= yellow" should be "N = blue, Br = yellow".
- e. In the line of 93, "TEM images" should be "TEM image".
- f. In the line of 98, "shows" should be "show".
- g. In the line of 139 ~ 140, "organic dyes, metal complex and inorganic nanostructures etc." should be "organic dyes, metal complex, and inorganic nanostructures etc.".
- h. In the line of 142, "thus further indicated" should be "thus further indicating". Otherwise, the sentence is wrong.
- i. In the line of 146, "is longer than" should be "is larger than".
- j. In the line of 157 and 158, " τ_1 " and " τ_2 " should be " τ_1 " and " τ_2 ", respectively.
- k. In the line of 257 and 258, "is reacted with" and "is recovered" should be changed into "reacted with" and "recovered", respectively.

Response to reviewers' comments

To reviewer #1:

1. The authors claim that the difference of amount of the perovskite NCs in MOF matrix estimated by EDX analysis of TEM (1%) and XPS data (19%) is due to the different detecting depths. However, the amounts from the two results are quite different. Could the authors give some statistic data? Since MABr can only react with a portion of MOFs, they should carefully consider the uniformity of the samples.

Response: Thank you for your suggestion. According to your advices, we did some additional experiments and analysis to make sure the accuracy of our data.

Firstly, the official description of the XPS analysis instrument (Kratos Axis Ultra^{DL}) we used– “*Efficient collection of photoelectrons by the magnetic and electrostatic lenses combined with the large 165 mm mean radius hemispherical analyzer ensure that the AXIS Supra boasts unrivalled sensitivity in large area analysis mode.*” (cited from the website: <http://www.kratos.com/products/axis-supra>), indicates that it has a relative large analysis area. Therefore we thought the XPS data from the analysis area should represent well the surface state of the whole sample. To further prove this opinion, we conducted the XPS characteristic three times. The results, shown in Supplementary Table 2, seem to be about the same, and the percentage of the MAPbBr₃ NCs in Pb-MOF estimated through the content of the Pb and Br elements are 19.1%, 20.2% and 23.4% (average value is 20.9%), thus indicating the reliability of this XPS data and the uniformity of the sample in relative large detection area.

Secondly, for the sake of more statistical EDX data, we initially tried to collect data from different area of MAPbBr₃ NCs@Pb-MOF particles under HRTEM as well. However, the obtained data in five detect areas fluctuated quite a lot. Comparatively speaking, MAPbBr₃ NCs@Pb-MOF particles are quite large (dozens of micrometers), and their surface are really rough, so the measured percentage of MAPbBr₃ NCs in Pb-MOF maybe varied with the detect locations. Suppose that the MAPbBr₃ NCs cover all the surface of MOF. When the detection area is located on the central and flat particle surface, the EDX signals are collected from the layer of central MAPbBr₃ NCs@Pb-MOF particle (I in Figure R1a). Another possible detection area is on the edge of MAPbBr₃ NCs@Pb-MOF, the EDX signals are mainly from the MAPbBr₃ NCs located on the edge of the particle (II in Figure R1a). It is obvious that the MAPbBr₃ NCs/MOF ratio on the central particle surface is much less than the edge. For the aforementioned reason, the obtained EDX data in different detect areas fluctuated significantly. Although the EDX analysis maybe a good way to check the chemical compositions for samples with good dispersity on TEM copper grids, such as CdSe and perovskite QDs which assembled to a single layer without big composition difference in different locations (III in Figure R1b), for our MAPbBr₃ NCs@Pb-MOF sample, the surface is so rough that the data cannot be very trustable to support our claim that the MAPbBr₃ NCs are mainly located in the outer part of MOF particles.

Figure R1 Schematic cross-section view of different detected areas (I , II and III) of the EDX analysis on (a) our MAPbBr₃ NCs@Pb-MOF particle sample and (b) assembled QDs single layer sample.

In order to select a powerful tool to simultaneously measure the content of Pb and Br with very large detecting depths and large analysis area, we finally focused on the X-Ray Fluorescence measurement (XRF-1800 spectrometer equipped with a Rh anode X-ray tube). Due to the large detecting depths and large analysis area, the data obtained from XRF can be representative for the whole sample. Therefore this method has been widely used to explore the chemical composition and element content in many industries and fields. As for our MAPbBr₃ NCs@Pb-MOF particle sample, we conducted this XRF measurement three times. As shown in Supplementary Table 1, there is no great difference between these results. Due to the light weight and the small amount of the N element in our sample, the content of this element has not been detected. (Tsuji, K. & Nakano, K. *X-Ray Spectrometry*, **36(3)**, 145-149 (2007)) Based on the contents of the Pb and Br, the percentage of the MAPbBr₃ NCs in Pb-MOF can be estimated as 2.73%, 3.76% and 2.95% with an average value of 3.15%. Obviously, it is much less than the XPS data (about 20.9%). Considering the different detecting depths of the XPS and XRF measurement, we can confirm that the MAPbBr₃ NCs are mainly located in the outer part of MOF crystals.

Action taken:

1. The EDX data has been removed from our original manuscript and the statistic XRF data has been added in Supplementary Information as Supplementary Table 1:

Supplementary Table 1 Element contents of the MAPbBr₃ NCs@Pb-MOF sample from the XRF data.

Sample	*Element	Wt %	At %	#Calculated percentage of MAPbBr ₃ NCs/Pb- MOF %	Average
1	Pb	47.75	5.49	2.73	
	C	37.4	74.24		
	O	13.31	19.82		
	Br	1.51	0.45		
	Pb	49.75	5.94		
2	C	36.95	76.21	3.76	3.15
	O	11.1	17.17		
	Br	2.16	0.67		
	Pb	59.02	8.36		
	C	31.91	78.05		
3	O	7	12.84	2.95	
	Br	2.03	0.74		

* Due to the light weight and the small amount of the N element in our sample, the content of this element has not been detected.

The percentage of the perovskite NCs in MOF matrix are estimated by the atomic percent of Pb and Br.

2. Statistic XPS data has been added in Supplementary Information as Supplementary Table 2:

Supplementary Table 2 Element contents of the MAPbBr₃ NCs@Pb-MOF sample from the XPS data.

Sample	Elements	Pos.	FWHM	Area	At %	#Calculated percentage of MAPbBr ₃ NCs/Pb- MOF %	Average
1	C 1s	284.84	1.66	5446.29	62.37	19.07	20.88
	N 1s	398.80	1.84	239.43	1.60		
	O 1s	528.60	2.57	6586.57	26.88		
	Pb 4f	136.00	1.61	15236.19	5.82		
	Br 3p	65.70	2.09	1108.65	3.33		
2	C 1s	284.80	2.60	8175.19	65.78	20.18	
	N 1s	401.30	1.91	267.69	1.26		
	O 1s	532.70	3.99	8652.08	24.81		

	Pb 4f	138.50	2.09	18893.46	5.07	
	Br 3p	68.80	2.61	1449.93	3.07	
	C 1s	284.80	1.50	9729.56	69.69	
	N 1s	401.90	1.61	790.77	2.06	
3	O 1s	531.40	2.27	7949.96	20.71	23.40
	Pb 4f	138.70	1.34	18549.00	4.43	
	Br 3p	68.20	1.96	1654.23	3.11	

The percentage of the perovskite NCs in MOF matrix are estimated by the atomic percent of Pb and Br.

3. The description of the amount of the perovskite NCs in MOF matrix in original manuscript: “In addition, the amount of the perovskite NCs in MOF matrix was estimated by EDX analysis of TEM (only approximate 1%, Supplementary Table 1) and XPS data (approximate 19%, Supplementary Table 2). Notably, the difference of the two results can be due to the different detecting depths^{38,39} of this two analysis methods, which may suggest that perovskite NCs are mainly located in the outer part of MOF crystals. However, based on the EDS data of our MAPbBr₃ NCs@Pb-MOF powder, the consumption of Pb element from Pb-MOF for constructing MAPbBr₃ during the first conversion process is very few (only ~ 1%).” has been changed in the current revised manuscript as follow: “Furthermore, the percentage of the perovskite NCs in MOF matrix was estimated by XRF (only approximate 3%, Supplementary Table 1) and XPS analysis (approximate 21%, Supplementary Table 2). Notably, the difference between these results can be ascribed to the different detecting depths of the two analysis methods. Compared with XRF, the detecting depth of XPS is only several nanometers,^{38,39} which may suggest that perovskite NCs are mainly located in the outer part of MOF particle.”

4. The description of the composition measurement in the “**Method**” section: “The surface composition was determined by the X-ray photoelectron spectroscopy (XPS, Kratos Axis Ultra^{DLD}), and all the binding energies were calibrated with the C 1s peak at 284.8 eV.” has been changed in the current revised manuscript as follow: “The chemical compositions were determined by the X-ray photoelectron spectroscopy (XPS, Kratos Axis Ultra^{DLD}, all the binding energies were calibrated with the C 1s peak at 284.8 eV) and X-Ray Fluorescence measurement (XRF-1800 spectrometer equipped with a Rh anode X-ray tube).”

2. In Figure 2c,d, the authors show MAPbX₃NCs@Pb-MOF can be tuned over the entire visible spectral region by adjusting their halide composition. What is the exact halide composition for each sample?

Response: Thank you for your advice. According to your suggestion, we numbered all this samples with different halide composition from 1 to 7. For clarity, we used the molar ratio of the added halide to represent the halide composition for each sample which has been listed in the caption of Figure 2.

Action taken: The number has been added in the Figure 2c, d, and the description of the halide composition for each sample also has been present in the caption. Figure 2 was redrew and presented as following:

Figure 2 | Optical properties of MAPbX₃ NCs@Pb-MOF. (a) PL emission spectra of Pb-MOF (black line) and MAPbBr₃ NCs@Pb-MOF (green line); (b) Time-resolved PL decay curve of MAPbBr₃ NCs@Pb-MOF detected at 527 nm with excitation of 450 nm; (c) Optical images under ambient light and 365 nm UV lamp and (d) PL emission spectra of MAPbX₃ NCs@Pb-MOFs (1: MAPbCl₃, 2: MAPbCl₂Br, 3: MAPbClBr₂, 4: MAPbBr₃, 5: MAPbBr₂I, 6: MAPbBrI₂, 7: MAPbI₃).

3. In the multiple information encryption and decryption processes, the authors found that the MAPbBr₃ NCs@Pb-MOF powder encounter obvious degradation. They guess the MAPbBr₃ NCs@Pb-MOF powder may degrade to some new compounds like MA₄PbBr₆·MeOH, MA_xPbBr_{2+x}, PbBr₂, Pb(OH)₂ or PbCO₃. I hope the authors give sufficient evidence for this assumption, such as Raman or FTIR, etc.

Response: Thank you for your suggestion. This is a good point. Accordingly, we tried our best to analyze the degradation products of MAPbBr₃ NCs@Pb-MOF in detail.

Firstly, we conducted the FTIR and XPS characterization for the MAPbBr₃ NCs@Pb-MOF powder before and after methanol impregnation. Similar with the XRD characterization, the IR signals of the MAPbBr₃ NCs are almost overlapped due to the present of the MOF matrix (Supplementary Fig. 19b). We can only identify the emergence and the disappearance of the new weak peak at 977 cm⁻¹, corresponding to the stretch modes of C–N from the MAPbBr₃, (Glaser, T. et al. *J. Phys. Chem. Lett.* **6**, 2913-2918 (2015).) after the conversion from the Pb-MOF and the subsequent methanol impregnation respectively. The XPS spectra of the MAPbBr₃ NCs@Pb-MOF powder sample before and after methanol impregnation has been shown in Supplementary Fig. 19c, d. Clearly, after degradation, the signal of Br significantly reduced, meanwhile the N species almost disappeared. These data all suggest that the luminescent MAPbBr₃ phase can be degraded by methanol treatment. Considering the small amount of the perovskite NCs and the complex hybrid system, it's difficult to determine the accurate degradation products from these results. In order to further figure out the degradation pathway, we directly did the methanol impregnation experiment of MAPbBr₃ bulk sample to avoid the influence of the MOF matrix (the Pb-MOF does not change or degrade in methanol). The XRD and FTIR characterization data are shown in Supplementary Fig. 20. Obviously, after methanol impregnation, the crystal structure of MAPbBr₃ has been changed to PbBr₂. Moreover, these C, N-containing groups have almost disappeared, which is consistent with the XPS data of MAPbBr₃ NCs@Pb-MOF sample. Therefore, we consider that after methanol impregnation, the MAPbBr₃ NCs have been mainly degraded to PbBr₂ and other C, N-containing organic constituents which can be easily washed away because of their good solubility (in methanol) and volatility.

The above results indicate that the degradation pathway of MAPbBr₃ in methanol is very different from the perovskite photovoltaic device in ambient atmosphere which we employed to speculate in our original manuscript. (Christians, J. A., Miranda Herrera, P. A. & Kamat, P.V. *J. Am. Chem. Soc.* **137**, 1530-1538 (2015), Leguy, A. M. A. *et al. Chem. Mater.* **27**, 3397-3407 (2015), Huang, W., Manser, J S., Kamat, P V. & Ptasinska, S. *Chem. Mater.* **28**, 303-311 (2015).) We thank the reviewer again very much for this constructive comment to strongly improve our work.

Action taken:

1. In the current manuscript, the description about the degradation of MAPbBr₃ NCs in Pb-MOF: “From the XRD patterns (Supplementary Fig. 15), during one cycle of impregnation-recovery, the distinct diffraction peak of at 14.9° disappear quickly and then recover again, ...the Pb element from the destroyed MAPbBr₃ NCs may play a relatively small role for the next conversion process.” has been changed as follow: “From the XRD and FTIR data (Supplementary Fig. 19a, b) of the MAPbBr₃ NCs@Pb-MOF powder sample after methanol impregnation, the distinct diffraction peak at 14.9° and the weak absorption peak at 977 cm⁻¹ both disappear quickly. Furthermore, the XPS characterization of the MAPbBr₃ NCs@Pb-MOF powder, shown in Supplementary Fig. 19c, d, clearly reveals that after degradation, the signal of Br significantly reduced, while the N species almost disappeared. These data all suggest that the luminescent MAPbBr₃ phase can be degraded by methanol treatment. Considering the small amount of the perovskite NCs and the complex hybrid system, it's difficult to determine the accurate degradation products from these results. In order to further figure

out the degradation pathway, the methanol impregnation experiment of MAPbBr₃ bulk sample has been directly conducted to avoid the influence of the MOF matrix (the Pb-MOF does not change or degrade in methanol). The results (including XRD and FTIR), shown in Supplementary Fig. 20, demonstrate that after methanol impregnation, the MAPbBr₃ NCs may mainly decompose into PbBr₂ and other C, N-containing organic constituents which can be easily washed away because of their good solubility (in methanol) and volatility. However, based on the small amount of MAPbBr₃ NCs in Pb-MOF, the Pb element from the destroyed MAPbBr₃ NCs may play a relatively small role for the next conversion process.”

2. The corresponding measurement method of the FTIR has been added in the section of “**Method**”, as shown below: “Fourier transform infrared (FTIR) spectra were measured by using a Nicolet 6700 spectrometer (USA).”

3. The FTIR and XPS data of the MAPbBr₃ NCs@Pb-MOF before and after methanol impregnation treatment have been added in the Supplementary Fig. 19, and the XRD and FTIR data of the MAPbBr₃ bulk sample before and after methanol impregnation treatment have been added in Supplementary Information as Supplementary Fig. 20.

Supplementary Figure 19 XRD (a), FTIR (b) and XPS (c, d) characterization of the MAPbBr₃ NCs@Pb-MOF before and after methanol impregnation treatment.

Supplementary Figure 20 XRD pattern (a) and FTIR spectra (b) of the MAPbBr₃ bulk before and after methanol impregnation treatment.

In order to further figure out the degradation pathway, we directly did the methanol impregnation experiment of MAPbBr₃ bulk sample to avoid the influence of the MOF matrix (the Pb-MOF does not change or degrade in methanol). Obviously, after methanol impregnation, the crystal structure of MAPbBr₃ has been changed to PbBr₂ (Supplementary Fig. 20a). Moreover, these C, N-containing groups have almost disappeared (Supplementary Fig. 20b). Therefore, we considered that after methanol impregnation, the MAPbBr₃ NCs in Pb-MOF have been mainly degraded to PbBr₂ and other C, N-containing organic constituents which can be easily washed away because of their good solubility (in methanol) and volatility.

4. A new reference has been added to the revised manuscript:

66. Glaser, T. *et al.* Infrared Spectroscopic Study of Vibrational Modes in Methylammonium Lead Halide Perovskites. *J. Phys. Chem. Lett.* **6**, 2913-2918 (2015).

To reviewer #2:

i) The authors need to provide informations of mechanical properties, through adhesion and adhesion experiments, of the printed and MAPbX₃ NCs@Pb-MOF materials printed;

Response: Thank you for your advice. The typical tape peel test was conducted using a commercial adhesion tape to qualitatively assess the mechanical properties of the printed Pb-MOF and MAPbBr₃ NCs@Pb-MOF materials. As shown in Supplementary Fig. 13, after the tape tests, there was no change on the position of the Pb-MOF and the encryption process. Similarly, the MAPbBr₃ NCs@Pb-MOF pattern can also be kept with high quality after tape test, and we didn't find evident fluorescent traces on the tested tape. Therefore, we can confirm that the printed Pb-MOF and MAPbBr₃ NCs@Pb-MOF materials in our work have excellent mechanical stability for security protection applications.

Action taken:

1. In the current manuscript, the description about the adhesion experiment is shown as: "The mechanical properties of the Pb-MOF and MAPbBr₃ NCs@Pb-MOF pattern were also qualitatively assessed through a typical tape peel test."^{53,54} As shown in Supplementary Fig. 13, after tape adhesion and peeling, the Pb-MOF and

MAPbBr₃ NCs@Pb-MOF pattern both can kept their high quality, which indicates that printed Pb-MOF and MAPbBr₃ NCs@Pb-MOF materials have excellent mechanical stability for security protection applications.”

2. The tape test results have been added in Supplementary Information as Supplementary Fig. 13.

Supplementary Figure 13 Tape test results of printed Pb-MOF (a) and MAPbBr₃ NCs@Pb-MOF (b) pattern.

3. New references have been added to the revised manuscript:

56. Ma, S., Liu, L., Bromberg, V. & Singler, T. J. Fabrication of highly electrically conducting fine patterns via substrate-independent inkjet printing of mussel-inspired organic nano-material. *J. Mater. Chem. C* **2**, 3885-3889 (2014).

57. Jeong, J. W. et al. High-resolution nanotransfer printing applicable to diverse surfaces via interface-targeted adhesion switching. *Nat. Commun.* **5**, 5387 (2014).

ii) If the Pb salt were printed in place of ink MOF, maintaining the concentrations of Pb and solvent, the encryption capability of would be maintained? The experiment to expose the Pb-MOF to MA-Br indicates only that the Pb ions of the MOF react as MA-Br, But does not eliminate the other hypothesis. The authors need to provide these information;

Response: Thank you for your advice. According to your suggestion, the Pb²⁺ ink was prepared and used for information encryption and decryption application. The Pb²⁺ ink composition was similar to the Pb-MOF ink except the absence of the H₃BTC linker. Subsequently, the parchment was used as the substrate for printing

process. As shown in Supplementary Fig. 12a, the printed pattern was invisible. After the conversion with MABr, the fluorescent MAPbBr₃ NCs could be formed. This can be ascribed to the confinement effect of the textured structure of the parchment substrate. But the fluorescence seems to be relative dim compared with the MAPbBr₃ NCs@Pb-MOF pattern. Importantly, the fluorescent pattern was very blurry, making it difficult to identify the printed information, which suggests that information decryption capability was limited and the Pb²⁺ ink is not suitable for the information protection application. Actually, this phenomenon is not unexpected given that the absence of organic linker. In this case, because of the absence of any H₃BTC linker, the printed Pb²⁺ pattern cannot be fixed by the crystal framework, possibly making them easy to move around when the parchment substrate was contact with high affinity solvent (such as alcohol). To further verify the feasibility of the Pb²⁺ ink, we used the PET substrate. The results were shown in Supplementary Fig. 12b, it is obvious that the fluorescent perovskite NCs cannot be formed in this case. This can also be attributed to the absence of organic linker and crystal framework, which cannot restrict or confine the growth of the perovskite NCs on the smooth flat PET surface. From these results, we can demonstrate that the Pb²⁺ ink cannot be directly used for the information encryption and decryption application in our work.

Action taken:

1. In the current manuscript, the description about the adhesion experiment is shown as: “To demonstrate the necessity and the role of the MOF structure in the security protection application of our platform, the corresponding Pb²⁺ ink (without H₃BTC linker) was prepared and used for information encryption and decryption process. As shown in Supplementary Fig. 12, it is obvious that the printed pattern using Pb²⁺ ink cannot maintain the information encryption and decryption capability on substrates.”

2. The results of the Pb²⁺ ink for the information encryption and decryption application have been added in the Supplementary Information as Supplementary Fig. 12.

Supplementary Figure 12 Photographs of the printed patterns on parchment (a) and PET (b) substrate using Pb²⁺ ink before and after MABr treatment under ambient and UV light.

*The Pb²⁺ ink composition was similar to the Pb-MOF ink except the absence of the H₃BTC linker. Subsequently, the parchment was used as the substrate for printing process. As shown in Supplementary Fig. 12a, the printed pattern was invisible. After the conversion with MABr, the fluorescent MAPbBr₃ NCs could be

formed. This can be ascribed to the confinement effect of the textured structure of the parchment substrate. But the fluorescence seems to be relative dim and blurry compared with the MAPbBr₃ NCs@Pb-MOF pattern, making it difficult to identify the printed information. Actually, this phenomenon is not unexpected given that the absence of organic linker. In this case, because of the absence of any H₃BTC linker, the printed Pb²⁺ pattern cannot be fixed by the crystal framework, possibly making them easy to move around when the parchment substrate was contact with high affinity solvent (such as alcohol). To further verify the feasibility of the Pb²⁺ ink, we used the PET substrate. The results were shown in Supplementary Fig. 12b, it is obvious that the fluorescent perovskite NCs cannot be formed in this case. This can also be attributed to the absence of organic linker and crystal framework, which cannot restrict or confine the growth of the perovskite NCs on the smooth flat PET surface. From these results, we can demonstrate that the Pb²⁺ ink cannot be directly used for the information encryption and decryption application in our work.

iii) The degradation of MOF seems to be a limiting factor for the application of this approach to explore the anti-counterfeiting applications. The limitations must be related in the manuscript;

Response: Thank you for your advice. According to your suggestion, the stability of the Pb-MOF powder and the Pb-MOF pattern were assessed through the thermogravimetry analysis (TGA) and XRD characterization. As shown in Supplementary Fig. 11a, the TGA curve indicates the good thermal stability (that is stable to 400 °C) of the as-synthesized Pb-MOF, which is consistent with previous report (Sadeghzadeh, H. & Morsali, A. *J. Coord. Chem.* **63**, 713-720 (2010)). Moreover, the XRD data of the Pb-MOF powder storage after 5 months (Supplementary Fig. 11b) shows no any change of the crystal structure. On the other hand, the XRD characterization of the printed Pb-MOF pattern storage after 3 months was also provided and shown in Supplementary Fig. 11c. Obviously, the diffraction peaks remains unchanged as well. All these results have powerfully demonstrated the excellent stability of the Pb-MOF for the application of security protection.

Action taken:

1. In the current manuscript, the description about the degradation of Pb-MOF is shown as: “For the security protection applications, the stability of MOF seems to be a critical factor. As shown in Supplementary Fig. 11a, the thermogravimetry analysis (TGA) shows that the as-synthesized Pb-MOF is stable to 400 °C indicating that it has a good thermal stability, which is consistent with previous report.³¹ On the other hand, from the XRD characterization (Supplementary Fig. 11b, c), it is obviously that both the Pb-MOF powder and the printed Pb-MOF pattern can remain original crystal structure after storage of several months, thus suggesting the excellent storage stability.”

2. The results of the degradation assessment of Pb-MOF have been added in the Supplementary Information as Supplementary Fig. 11.

Supplementary Figure 11 Stability of the Pb-MOF powder and the Pb-MOF pattern: a) TG curve of the Pb-MOF powder, b) XRD patterns of the Pb-MOF powder before and after storage 5 months, c) XRD data of the printed Pb-MOF pattern after storage 3 months.

3. The description of the TGA measurement has been added in the “**Method**” section as follow:

“Thermogravimetry analysis (TGA) was conducted on a Mettler Toledo analyzer from 30 °C to 700 °C at a heating rate of 10 °C min⁻¹ with an Ar flow rate of 50 mL min⁻¹.”

iv) Provide adequate section title;

Response & Action taken: Thank you. These two section titles—“**Morphology and structure characterization.**” and “**Universality of perovskite NCs- MOF platform**”—have been added in the proper position of our current revised manuscript.

v) Improve the legends of Figures 2 and 4, in order to facilitate the correlation with the materials;

Response: Thank you for your advice. We numbered all this powder samples (from 1 to 7 for Figure 2; from 1 to 3 for Figure 4), and listed them in the corresponding caption.

Action taken: Figure 2 and Figure 4 were redraw and presented as following:

Figure 2 | Optical properties of MAPbX₃ NCs@Pb-MOF. (a) PL emission spectra of Pb-MOF (black line) and MAPbBr₃ NCs@Pb-MOF (green line); (b) Time-resolved PL decay curve of MAPbBr₃ NCs@Pb-MOF detected at 527 nm with excitation of 450 nm; (c) Optical images under ambient light and 365 nm UV lamp and (d) PL emission spectra of MAPbX₃ NCs@Pb-MOFs (1: MAPbCl₃, 2: MAPbCl₂Br, 3: MAPbClBr₂, 4: MAPbBr₃, 5: MAPbBr₂I, 6: MAPbBrI₂, 7: MAPbI₃).

Figure 4 | The reversible on/off switch property of perovskite NCs-MOF platform. (a) Sequential optical images and PL emission spectra of MAPbBr₃ NCs@Pb-MOF after one cycles of impregnation-recovery process under ambient light. **1, 2, 3** represent the original, impregnated, and recovered powder sample of MAPbBr₃ NCs@Pb-MOF respectively; (b) PL intensity, peak wavelength and FWHM of MAPbBr₃ NCs@Pb-MOF in the impregnation-recovery cycles as a function of cycle number; (c) Reversible fluorescence switching of the MAPbBr₃ NCs@Pb-MOF pattern in one encryption-decryption cycle (methanol impregnation for encryption and MABr spraying for decryption).

vi) Improve discussion of the Figure 2;

Response: Thank you very much. We carefully checked the discussion of Figure 2 and added some detail description about the PLQY and PL lifetime results.

Action taken:

1. For better description of PLQY result, the sentences: “Compared to the luminescent perovskite NCs synthesized by conventional solution-processable strategies initially outlined by Pérez-Prieto et al,²⁰ the relatively lower PLQYs of the MAPbBr₃ NCs@Pb-MOF can be attributed to the absence of any surface shelling and insufficient ligand passivation. In spite of this, it is comparable to or even brighter than these reported MAPbBr₃ NCs confined synthesized in porous matrix,³²⁻³⁶ and sufficient for information identification applications.” have been added in proper position of the “**Optical characterization**” section.

2. As for the detail and accurate discussion of the result of PL lifetime in the “**Optical characterization**” section, the sentence : “Similar to the previous report,³⁵ the shorter lifetime could be the result of dominant surface trapping of the MAPbBr₃ NCs, and the longer lifetime is attributed to the recombination of generated excitons upon light absorption.” has been changed into “Similar to the previous report,³⁵ the shorter lifetime could be the result of dominant surface trapping of the MAPbBr₃ NCs, suggesting that the non-radiative recombination pathway has non-negligible contribution in our MAPbBr₃ NCs@Pb-MOF sample, which is consistent with the abovementioned result of relative low PLQY.”

vii) *The sections name need be reviewed, specifically the introduction.*

Response & Action taken: Thank you. We carefully reviewed the sections names. For the sake of simplicity and clearly, the section names of “**The conversion process of Pb-MOF to the MAPbBr₃ NCs@Pb-MOF**” and “**Optical characterization of MAPbX₃ NCs@Pb-MOF powder**” have been changed as: “**Conversion process**” and “**Optical characterization**”.

To reviewer #3:

(1) The authors have emphasized luminescent perovskite NCs obtained from the conversion of the invisible Pb-MOF as an important potential application for confidential information encryption and decryption. Even though the stability of the perovskite NCs has been improved by the protection effect of the textured substrate and the MOF matrix, it seem still incompatible with the practical application in view of their toxic nature of the Pb²⁺ ion in the perovskite NCs. The authors are suggested to address this concern with more discussion or perspectives to tackle this technical challenge.

Response: Thank you for your kind suggestion. We agree with the reviewer that the toxic nature of the Pb²⁺ ion in the perovskite NCs could be an issue for potential applications in security protection. We have carefully evaluated this issue and proposed the following possible approach to circumvent this technical challenge—exploring the toxic metal free perovskite materials. According to a great amount of fascinating research

progresses of lead-free perovskite materials for photonic and optoelectronic applications, we considered that exploring the toxic (or lead) metal free perovskite materials would become one of the most promising approaches in our platform. Typically, Jellicoe et al. and Wang et al. have both reported successful replacement of lead with non-toxic tin and synthesizing CsSnX₃ and Cs₂SnI₆ NCs respectively, which open up new opportunities to explore new types of lead-free perovskite materials. (Jellicoe, T. C. *et al. J. Am. Chem. Soc.* **138**, 2941-2944 (2016), Wang, A. *et al. Chem. Mater.* **28**, 8132-8140 (2016)) In addition, Leng and co-workers prepared a novel MA₃Bi₂Br₉ QDs with high PLQY, suggesting that bismuth is another promising choice to constitute toxic metal free perovskite materials. (Leng, M. *et al. Angew. Chem. Int. Edit.* **55**, 15012-15016 (2016))

Action taken:

1. In the current revised manuscript, the detail discussion and perspectives about the toxic issue of the Pb²⁺ ion in the perovskite NCs has been improved and shown as: “It should be mentioned that the toxicity of Pb in lead halide perovskite⁶⁷⁻⁶⁹ is a considerable concern for future applications of our platform. However, we think it won't be a fatal problem according to the recent fascinating research progresses in the synthetic chemistry of lead-free perovskite materials. Typically, Jellicoe et al. and Wang et al. have both reported successful replacement of lead with non-toxic tin and synthesizing CsSnX₃ and Cs₂SnI₆ NCs respectively.^{70,71} In addition, Leng and co-workers prepared a novel MA₃Bi₂Br₉ QDs with high PLQY, suggesting that bismuth is another promising choice to constitute toxic metal free perovskite materials.⁷² Anyway we believe that this strategy will open up a potential avenue for luminescent perovskite materials in security protecting applications.”

2. New references 70-72 were added to the manuscript:

70. Jellicoe, T. C. *et al. Synthesis and Optical Properties of Lead-Free Cesium Tin Halide Perovskite Nanocrystals. J. Am. Chem. Soc.* **138**, 2941-2944 (2016).

71. Wang, A. *et al. Controlled Synthesis of Lead-Free and Stable Perovskite Derivative Cs₂SnI₆ Nanocrystals via a Facile Hot-Injection Process. Chem. Mater.* **28**, 8132-8140 (2016).

72. Leng, M. *et al. Lead-Free, Blue Emitting Bismuth Halide Perovskite Quantum Dots. Angew. Chem. Int. Edit.* **55**, 15012-15016 (2016).

(2) In the section of “Optical characterization of MAPbX₃ NCs@Pb-MOF powder”, the author considered that the Pb-MOF powder, showing white color (Figure 1b), is still visible under ambient light due to the existing scattering phenomenon. Then, in the section of “Confidential information encryption and decryption application”, the authors claimed that the printed Pb-MOF pattern is indeed invisible absolutely because of the reduced or even negligible scattering phenomenon. This point is not enough clear, because the substrates used

for the printing process seem also white color, it means you are printing white MOFs on a white substrate, which may confuse the observation, The author should give some references to support the standpoints of the existing scattering phenomenon of MOF powder and the reduced or even negligible scattering phenomenon through reduce the crystal size of MOF.

Response: Thank you for your advice. This is a very good point. In this case, because of the semitransparent and white color of the used parchment substrate, the direct observation of the reduced scattering of Pb-MOF nanocrystals would be confused. However, from the photographs of the invisible Pb-MOF patterns on transparent PET substrate (Figure R2), we can powerful confirm the reduced or even negligible scattering phenomenon due to the reduced crystal size. On the basic of the classical scattering theory, (van de Hulst, H. C. *Light Scattering by Small Particles*, Wiley, New York 1957, Wolf, P. E & Maret, G. *Phys. Rev. Lett.* **55**, 2696-2699 (1985), Chen, D., Huang, F., Cheng, Y. B. & Caruso, R. A. *Adv. Mater.* **21**, 2206-2210 (2009)) the scattering of light by particles highly depends on the particle size and wavelength of the incident radiation. Similarly, because of the negligible light scattering, some wide band gap semiconductor nanocrystals (such as TiO₂) have been widely used as transparency charge transport layer in optoelectronic devices. (Chen, D., Huang, F., Cheng, Y. B. & Caruso, R. A. *Adv. Mater.* **21**, 2206-2210 (2009))

Figure R2 Invisible Pb-MOF pattern on transparent PET substrate.

Action taken:

New references 53-55 were added to the manuscript:

53. van de Hulst, H. C. *Light Scattering by Small Particles*, Wiley, New York 1957

54. Wolf, P. E & Maret, G. Weak localization and coherent backscattering of photons in disordered media. *Phys. Rev. Lett.* **55**, 2696-2699 (1985).

55. Chen, D., Huang, F., Cheng, Y. B. & Caruso, R. A. Mesoporous anatase TiO₂ beads with high surface areas and controllable pore sizes: a superior candidate for high-performance dye-sensitized solar cells. *Adv. Mater.* **21**, 2206-2210 (2009).

(3) More details should be added in certain Figures' caption, for example, what are the emission and excitation wavelengths of the decay spectrum in the time-resolved decay curve measurement in Fig. 2b; what the 7 samples in Fig. 2c and 2d are; how to treat the printed substrate by MeOH and MABr in Fig. 4c.

Response & Action taken:

1. The emission (527 nm) and excitation (450 nm) wavelengths have been added in the caption of the Figure 2b.
2. The number has been added in the Figure 2c, d, and the description of these samples with different added halide composition also has been present in the caption.

Figure 2 | Optical properties of MAPbX₃ NCs@Pb-MOF. (a) PL emission spectra of Pb-MOF (black line) and MAPbBr₃ NCs@Pb-MOF (green line); (b) Time-resolved PL decay curve of MAPbBr₃ NCs@Pb-MOF detected at 527 nm with excitation of 450 nm; (c) Optical images under ambient light and 365 nm UV lamp and (d) PL emission spectra of MAPbX₃ NCs@Pb-MOFs (1: MAPbCl₃, 2: MAPbCl₂Br, 3: MAPbClBr₂, 4: MAPbBr₃, 5: MAPbBr₂I, 6: MAPbBrI₂, 7: MAPbI₃).

3. The description of the treatment method of the printed substrate by MeOH and MABr has been present in the caption of Figure 4c as follow:

Figure 4 | The reversible on/off switch property of perovskite NCs-MOF platform. (a) Sequential optical images and PL emission spectra of MAPbBr₃ NCs@Pb-MOF after one cycles of impregnation-recovery process under ambient light. 1, 2, 3 represent the original, impregnated, and recovered powder sample of MAPbBr₃ NCs@Pb-MOF respectively; (b) PL intensity, peak wavelength and FWHM of MAPbBr₃ NCs@Pb-MOF in the impregnation-recovery cycles as a function of cycle number; (c) Reversible fluorescence switching of the MAPbBr₃ NCs@Pb-MOF pattern in one encryption-decryption cycle (methanol impregnation for encryption and MABr spraying for decryption).

(4)Abbreviations should be clarified when they are first used, for example, MABr is a critical reagent in this study, but I did not get what it is.

Response: Thank you. The abbreviations of MAPbBr₃ and MABr have been clarified as CH₃NH₂PbBr₃ and CH₃NH₂Br when they are used first respectively.

Action taken:

1. The detail descriptions of the abbreviations of MAPbBr₃: “CH₃NH₂PbBr₃ (MAPbBr₃)” have been added.
2. The detail descriptions of the abbreviations of MABr: “CH₃NH₂Br (MABr)” have been added.

(5)The emission spectra of MAPbBr₃NCs@Pb-MOF under different excitation wavelengths should be measured.

Response: Thank you. The excitation-emission matrix (EEM) spectrum has been conducted to reveal the integrated PL information.

Action taken:

1. The EEM and corresponding caption have been added in Supplementary Information as Supplementary Fig. 6.

Supplementary Figure 6 EEM spectrum of the MAPbBr₃ NCs@Pb-MOF powder.

2. In the current manuscript, the description is shown as: “The excitation-emission matrix (EEM) spectrum of the MAPbBr₃ NCs@Pb-MOF powder, shown in Supplementary Fig. 6, reveals that the PL emission peak is not excitation wavelength dependent.”

3. The corresponding measurement method of the EEM spectrum has been added in the section of “**Method**”, as shown below: “The total luminescence spectra were characterized in the form of excitation-emission matrix (EEM) (F-7000, Hitachi) with the scanning emission spectra varied from 450 to 600 nm by increasing the excitation wavelength from 300 to 550 nm at 5 nm increments.”

(6) High resolution TEM and FFT images of the “small MAPbBr₃ QDs” should be applied to prove the same crystal structure with MAPbBr₃ NCs in Pb-MOF.

Response: Thank you for your advice. According to your suggestion, the high resolution TEM and FFT image of one individual MAPbBr₃ QDs have been added in Supplementary Fig. 8. The interplanar spacing of about 2.9 Å corresponding to the (200) crystal faces of the MAPbBr₃ crystal can be easily confirmed.³⁷

Action taken:

1. The high resolution TEM and FFT image of one individual MAPbBr₃ QDs have been added in Supplementary Fig.8c.

2. The description of the high resolution TEM and FFT image of one individual MAPbBr₃ QDs has been added as follow: “from which the interplanar spacing of about 2.9 Å corresponding to the (200) crystal faces of the MAPbBr₃ crystal can be confirmed.³⁷” In addition, for clarity of the revised manuscript, the whole description of “quantum confinement phenomenon” has been moved into the Supplementary Information.

The revised Supplementary Fig. 8 is shown as follow:

Supplementary Figure 8 TEM images (a, b), HRTEM image (c), and size distribution (d) of the MAPbBr₃ NCs@Pb-MOF sample without drying step.

To prove the quantum confinement phenomenon during the perovskite NCs' growth process, we conducted the TEM characterization of the powder sample before rinsing and drying. As shown in Supplementary Fig. 8, the small MAPbBr₃ QDs (~ 3.3 nm) are embedded in Pb-MOF matrix with good dispersion, from which the interplanar spacing of about 2.9 Å corresponding to the (200) crystal faces of the MAPbBr₃ crystal can be confirmed.³⁷

(7)As to the encryption and decryption processes, the authors stated that “for the further encryption process, the substrate with patterns was immersed in methanol solution for about 10 min, the green emission was gradually

quenched and finally disappeared absolutely”, I think that the printed materials on the substrate should gradually dissolve in methanol, because the printed material is also dissolved in alcohols solution (i.e. ethanol and ethylene glycol).

Response: Thank you for your comment. We agree with the reviewer that the dissolving or degradation of printed Pb-MOF pattern in methanol is critical for the encryption and decryption application. In this case, we hold that the printed Pb-MOF pattern on substrate has excellent stability in methanol. We should mention that in our stable and security ink the solvent for Pb²⁺ and H₃BTC linker is DMSO, while ethanol and ethylene glycol only act as solvent combination. The ethanol plays the role of precipitating agents which promotes the nucleation reaction of MOF crystals and the presence of ethylene glycol can dramatically improve the viscosity and stability of the precursor solution. (Zhuang, J. -L., Ar, D., Yu, X. -J., Liu, J. -X. & Terfort, A. *Adv. Mater.* **25**, 4631-4635 (2013).) Therefore, the alcohols solution (i.e. ethanol and ethylene glycol) is not the agent for dissolving the printed material (Pb-MOF crystals). For powerful proving our opinion, the XRD characterization of Pb-MOF pattern after methanol impregnation has been conducted. As shown in Figure R3, after methanol impregnation for 30 min, the Pb-MOF pattern can completely remain the original crystal structure, which demonstrate that the printed Pb-MOF pattern cannot be dissolved in methanol and show excellent stability in multiple information encryption and decryption processes.

Figure R3 XRD characterization of Pb-MOF pattern after methanol impregnation

(8) *There are some sloppy statement and spelling mistakes in this manuscript. I listed out some of them below.*

- a. In the line of 77 ~ 78, “the emission color ... from white to blue green then to yellow green” should change into “the emission color ... from non-fluorescence to blue green then to yellow green” because the white color of MOF powder is not the emission color.
- b. In the line of 17, “quickly simply” should be “quickly and simply”.
- c. In the line of 54, “could” should be “can”.
- d. In the line of 86, “N=blue, Br= yellow” should be “N = blue, Br = yellow”.
- e. In the line of 93, “TEM images” should be “TEM image”.
- f. In the line of 98, “shows” should be “show”.
- g. In the line of 139 ~ 140, “organic dyes, metal complex and inorganic nanostructures etc.” should be “organic dyes, metal complex, and inorganic nanostructures etc.”.
- h. In the line of 142, “thus further indicated” should be “thus further indicating”. Otherwise, the sentence is wrong.
- i. In the line of 146, “is longer than” should be “is larger than”.
- j. In the line of 157 and 158, “ τ_1 ” and “ τ_2 ” should be “ τ_1 ” and “ τ_2 ”, respectively.
- k. In the line of 257 and 258, “is reacted with” and “is recovered” should be changed into “reacted with” and “recovered”, respectively.

Response & Action taken: Thank you very much for these kind reminding. These mistakes have been corrected in our current revised manuscript. In addition, we carefully checked the text and corrected other missing again. The amendments in the manuscript were marked in blue.

Reviewers' comments:

Reviewer #1 (Remarks to the Author):

In this manuscript, L. Li and co-workers reported a strategy to realize confidential information protection and storage based on the conversion of Pb based MOFs to luminescent perovskite NCs. It has demonstrated that the platform provided by authors can act as a smart luminescent system towards the confidential information encryption and decryption with various high-resolution patterns by inkjet-printing technique. It is of great importance to develop novel strategy for data security. In this sense, it seems like this paper is publishable after major revision.

Comments:

1. The information recording process is quite complicated because of high boiling point of ink (DMSO and EG). Can the authors find an alternative solvent as the ink?
2. Figure 2c showed different MAPbX₃ NCs@Pb-MOFs with various emission colors, but the synthetic detail is missing in the experimental section. Have the crystal structures of MAPbX₃ NCs@Pb-MOFs been confirmed?
3. Figure 4b indicated the PL intensity change of MAPbBr₃ NCs@Pb-MOF in the impregnation-recovery cycles. This reversible experiment should be carried out on the printed paper.
4. The authors claimed that the printed images have excellent resolution. Can the authors printed microscale patterns through their ink-jet printing system?

Reviewer #2 (Remarks to the Author):

I am satisfied with the author's responses and manuscript corrections.
I recommend accepting the manuscript.

Reviewer #3 (Remarks to the Author):

The authors had addressed my concerns for this manuscript properly, so I would like to suggest its acceptance to publication at this stage.

Response to reviewers' comment

To reviewer #1:

1. The information recording process is quite complicated because of high boiling point of ink (DMSO and EG). Can the authors find an alternative solvent as the ink?

Response: Thank you for your suggestion. This is a very good point. However, at present stage, we cannot find an alternative solvent (with lower boiling point) as the ink. And we found that these solvents with high boiling point (DMSO and EG) in the ink system don't affect our simple information encryption and decryption process.

In our printing process, the DMSO and EG play critical roles. As we know, in order to employ MOFs to record information via inkjet-printing process, we must obtain a clear and stable MOF precursor as ink with proper viscosity and surface tension. Therefore, firstly the solvent we used must simultaneously have good solubility of the respective starting materials (including metal salts and organic linkers). Based on this consideration, compared to common polar solvent (such as alcohol or water), DMSO (or DMF, but DMSO is better) has a stronger tendency to coordinate metal ions to make clear MOF precursor solution (Figure R1) (Ameloot, R. *et al. Adv. Mater.* **22**, 2685-2688 (2010)). Secondly, for inkjet-printing process, the viscosity and surface tension of the ink is critical as well. According to Zhuang's report (Zhuang, J. -L., Ar, D., Yu, X. -J., Liu, J. -X. & Terfort, A. *Adv. Mater.* **25**, 4631-4635 (2013)), EG we used actually act as a solvent combination to adjust the viscosity and surface tension of the Pb-MOF ink. Unfortunately, these important constituent both have relative high boiling point.

Figure R1 the Pb-MOF ink prepared by various solvent.

Because of the high boiling point of these solvents, the direct influence seems to be that these solvent residues are difficult to be removed from the printed MOF structure. For various applications of MOF films or patterns (such as sensor, gas capture, catalysis or molecular sieving), it is very important to remove residual solvents from MOFs completely, because the presence of them is very detrimental to MOF pore structure and thus affect their performance. So to solve this problem, according to the literature's report (Zhuang, J. -L., Ar, D., Yu, X. -J., Liu, J. -X. & Terfort, A. *Adv. Mater.* **25**, 4631-4635 (2013)), after printing, a solvent

development step (immersing printed MOF patterns into methanol solution) have been introduced to help to remove the less volatile solvents, which may be the main reason for the quite complicated information recording process considered by the reviewer. However, in our case, we found that the solvent development step does not seem to be the necessary step. In other word, even if the DMSO and EG still exist in our printed MOF structures, the information encryption and decryption application can also be realized with high quality (Figure R2). The reason may lie in that during the encryption and decryption process, the solvent used for conversion reaction (butanol) and the luminescent quenching (methanol) both contribute to the removing of these less volatile solvents, thus making the solvent development step not necessary. Therefore, even this ink system (containing solvents with high boiling point) is been used directly, it don't have distinct effect on our simple information encryption and decryption process.

Figure R2 Photographs of the printed patterns (a) without and (b) with solvent development step (methanol immersing treatment) before and after MABr treatment under ambient and UV light.

2. Figure 2c showed different MAPbX₃ NCs@Pb-MOFs with various emission colors, but the synthetic detail is missing in the experimental section. Have the crystal structures of MAPbX₃ NCs@Pb-MOFs been confirmed?

Response: According to your advice, we have added the synthetic detail of the mixed halide perovskite NCs samples in the “Method” section and confirmed their crystal structure via XRD characterization. As shown in Supplementary Information as Supplementary Fig. 10, the peaks of the (100) and (210) reflection gradually shift toward higher angles with smaller halide ions (Br, Cl) due to the reduced lattice parameters, which confirm the cubic perovskite phase for all single and mix-halide MAPbX₃ NCs samples. For the iodide-containing samples, the intensities of these peaks were relative lower compared with others. This may be ascribed to the fast degradation of the iodide-containing perovskite crystals in air.

Action taken:

1. The synthetic detail description of the MAPbX₃ NCs@Pb-MOFs samples has been added in the “Methods” section. “For the synthesis of MAPbX₃ NCs@Pb-MOF, ... then dried at 80 °C for 5 min before further characterization.” has been changed in the current revised manuscript as follow: “For the typical synthesis of MAPbBr₃ NCs@Pb-MOF, the Pb-MOF (200 mg) was dispersed in hexane (10 mL) with stirring at room temperature. Then 500 μL MABr/n-butanol solution (10 mg/mL) was added to the Pb-MOF/hexane suspension. After 30 s, the colored powder was collected by filtration and rinsed with n-butanol then dried at 80 °C for 5 min before further characterization. MAPbBr_xCl_{3-x} NCs@Pb-MOF and MAPbBr_xI_{3-x} NCs@Pb-MOF were fabricated via the similar strategy with the variation of the MAX/n-butanol solution constitution.”

2. In the current manuscript, the description about the crystal structures of MAPbX₃ NCs@Pb-MOFs is shown as: “From the XRD characterization (Supplementary Fig. 10), the peaks of the (100) and (210) reflection gradually shift toward higher angles with smaller halide ions (Br, Cl) due to the reduced lattice parameters, which confirm the cubic perovskite phase for all MAPbX₃ NCs samples.”

2. The XRD results of the MAPbX₃ NCs@Pb-MOF powder samples with different halide constitution have been added in Supplementary Information as Supplementary Fig. 10.

Supplementary Figure 10 XRD patterns of single and mix-halide MAPbX₃ NCs@Pb-MOF samples.

The peak intensities of these iodide-containing samples were relative lower compared with others, which may be ascribed to the fast degradation of the iodide-containing perovskite crystals in air.

3. Figure 4b indicated the PL intensity change of MAPbBr₃ NCs@Pb-MOF in the impregnation-recovery cycles. This reversible experiment should be carried out on the printed paper.

Response: Thanks for your kind suggestion. According to your comment, the reversible experiment on printed substrate has been conducted to confirm the multiple information encryption and decryption processes of our perovskite NCs-MOFs platform.

Action taken:

1. The reversible fluorescence switching of printed information on parchment paper and corresponding caption have been added in Supplementary Information as Supplementary Fig. 22.

Supplementary Figure 22 Reversible fluorescence switching of the MAPbBr₃ NCs@Pb-MOF pattern. (a) Sequential optical images and PL emission spectra of MAPbBr₃ NCs@Pb-MOF after two cycles of impregnation-recovery process under ambient light. 1, 2, 3, 4, 5 represent the original, impregnated, recovered, secondary impregnated and secondary recovered MAPbBr₃ NCs@Pb-MOF pattern respectively. (methanol impregnation for encryption and MABr spraying for decryption); (b) PL intensity of MAPbBr₃ NCs@Pb-MOF as a function of cycle number.

2. In the current manuscript, the description is shown as: “Moreover, even after 10 consecutive switching cycles, the quality of the information encryption and decryption process still remains unaffected (Supplementary Fig. 22).”

4. The authors claimed that the printed images have excellent resolution. Can the authors printed microscale patterns through their ink-jet printing system ?

Response: Thank you for your advice. This is a very good point. According to your suggestion, we try to print a microscale pattern (a thin line) on parchment through our ink-jet printing system. The fluorescent image, shown in Figure R3, indicates an attainable resolution better than 400 μm , which is comparable to the literature’s report with similar ink constitution (Zhuang, J. -L., Ar, D., Yu, X. -J., Liu, J. -X. & Terfort, A. *Adv. Mater.* **25**, 4631-4635 (2013)). Even though, we think that this printing technique seems to be sufficient for our purposes in the application of information protection, however, this resolution remains not good enough compared with other optoelectronic applications (Choi, M. K. *et al. Nat. Commun.* **6**, 7149-7156 (2015)). This can be ascribed to the size of the inkjet nozzle of the printer we used (*HP Desk Jet 2132*) and the good affinity of the paper substrate toward the Pb-MOF ink which easily infiltrate around (see in Figure R3). Based on this consideration, we changed the word “excellent” into “good” for more rigorous expression.

Figure R3 Fluorescent microscopy image of the microscale pattern (a thin line) on parchment. Scale bar: 200 μm .

Action taken: In the second paragraph of the “Confidential information encryption and decryption application” section, “with excellent resolution” has been changed into “with good resolution”.

REVIEWERS' COMMENTS:

Reviewer #1 (Remarks to the Author):

I think that this revised manuscript is suitable for publication without further revision.